# Tumor suppressor PALB2 maintains redox and mitochondrial homeostasis in the brain and cooperates with ATG7/autophagy to suppress neurodegeneration

Yanying Huo[1,2], Akshada Sawant[1,3], Yongmei Tan[4], Amar H Mahdi[1,2¤a], Tao Li[1,2], Hui Ma[1,2], Vrushank Bhatt[1,5], Run Yan[6,7¤b], Jake Coleman[1,8], Cheryl F Dreyfus[9], Jessie Yanxiang Guo[1,5,10], M. Maral Mouradian[6,7], Eileen White[1,3]*, Bing Xia[1,2]*

1 Rutgers Cancer Institute of New Jersey, New Brunswick, New Jersey, United States of America,
2 Department of Radiation Oncology, Rutgers Robert Wood Johnson Medical School, New Brunswick, New Jersey, United States of America, 3 Department of Molecular Biology and Biochemistry, Rutgers, The State University of New Jersey, Piscataway, New Jersey, United States of America, 4 Stomatological Hospital of Guangzhou Medical University, Guangzhou, P.R. China, 5 Department of Medicine, Rutgers Robert Wood Johnson Medical School, New Brunswick, New Jersey, United States of America, 6 Department of Neurology, Rutgers Robert Wood Johnson Medical School, New Brunswick, New Jersey, United States of America, 7 RWJMS Institute for Neurological Therapeutics, Rutgers Robert Wood Johnson Medical School, New Brunswick, New Jersey, United States of America, 8 Department of Neuroscience and Cell Biology, Rutgers Robert Wood Johnson Medical School, New Brunswick, New Jersey, United States of America, 9 Rutgers School of Environmental and Biological Sciences, New Brunswick, New Jersey, United States of America, 10 Department of Chemical Biology, Rutgers Ernest Mario School of Pharmacy, Piscataway, New Jersey, United States of America

¤a Current address: Department of Physiology, College of Medicine, Al-Mustansiriyah University, Baghdad, Iraq
¤b Current address: Sanyou Biopharmaceuticals, Shanghai, China.
* epwhite@cinj.rutgers.edu (EW); xiabi@cinj.rutgers.edu (BX)

**Data Availability Statement:** All relevant data are within the manuscript and its Supporting Information files.

## Abstract

The PALB2 tumor suppressor plays key roles in DNA repair and has been implicated in redox homeostasis. Autophagy maintains mitochondrial quality, mitigates oxidative stress and suppresses neurodegeneration. Here we show that *Palb2* deletion in the mouse brain leads to mild motor deficits and that co-deletion of *Palb2* with the essential autophagy gene *Atg7* accelerates and exacerbates neurodegeneration induced by ATG7 loss. *Palb2* deletion leads to elevated DNA damage, oxidative stress and mitochondrial markers, especially in Purkinje cells, and co-deletion of *Palb2* and *Atg7* results in accelerated Purkinje cell loss. Further analyses suggest that the accelerated Purkinje cell loss and severe neurodegeneration in the double deletion mice are due to excessive oxidative stress and mitochondrial dysfunction, rather than DNA damage, and partially dependent on p53 activity. Our studies uncover a role of PALB2 in mitochondrial homeostasis and a cooperation between PALB2 and ATG7/autophagy in maintaining redox and mitochondrial homeostasis essential for neuronal survival.

**Funding:** This work was supported by the National Cancer Institute grants R01CA188096 (BX and EW) and R01CA138804 (BX). MMM is the William Dow Lovett Professor of Neurology and is supported by NIH grants NS101134, NS096032 and NS116921, the Michael J. Fox Foundation for Parkinson's Research and the American Parkinson Disease Association. The funders had no role in study design, data collection and analysis, decision to publish, or preparation of the manuscript.

**Competing interests:** I have read the journal's policy and the authors of this manuscript have the following competing interests: EW is a founder of Vescor Therapeutics and a stockholder of Forma Therapeutics. MMM holds patents and is a founder of MentiNova, Inc. None of the above relates to the subject matter of this manuscript.

## Author summary

PALB2 is a tumor suppressor in which inherited mutations increase the risk of breast, ovarian, pancreatic, and other cancers. It plays a critical role in DNA repair and promotes antioxidant gene expression. ATG7 is an essential factor for autophagy, an intracellular waste disposal and nutrient recycling process. Loss of autophagy function leads to accumulation of toxic wastes and damaged mitochondria, leading to oxidative stress and other problems in the cell. As neurons in the brain are particularly sensitive to oxidative stress and waste accumulation, loss of ATG7 or autophagy in the brain causes death of neurons and neurodegeneration. In this study, we found that loss of PALB2 in the brain led to oxidative stress accompanied by increased amount of functionally impaired mitochondria, and that combined loss of PALB2 and ATG7 caused accelerated and more severe neurodegeneration than did ATG7 loss alone. We further found that the exacerbated phenotype was mainly caused by excessive oxidative stress, rather than increased DNA damage. Our studies establish a new function of PALB2, i.e., mitochondrial regulation, provide additional insights into the function of ATG7 in the brain, and further underscore the role of oxidative stress in neuronal death and neurodegeneration.

## Introduction

Neurodegenerative disorders affect approximately 15% of the population [1]. Oxidative stress, which occurs when production of reactive oxygen species (ROS) in the cell exceeds the capacity of its antioxidant system, has been recognized as a major contributing factor in neurodegenerative diseases. ROS are primarily byproducts of aerobic metabolism during mitochondrial energy production [2]. ROS overproduction can lead to oxidative damage to both cellular and mitochondrial proteins, DNA, and membrane lipids, impairing a range of cellular functions, including the ability of mitochondria to synthesize ATP and carry out other metabolic functions [3]. The brain is particularly susceptible to oxidative stress due to its high levels of polyunsaturated fatty acids, which are prone to oxidation, and its relatively high metabolic rate and high oxygen demand [4,5], strict aerobic metabolism [6] and low levels of antioxidants [7]. ROS-induced damage to lipids, proteins and DNA is a common feature of all major neurodegenerative disorders [8,9].

One cellular mechanism that protects against neurodegeneration is autophagy. Autophagy is a cellular waste disposal and nutrient recycling mechanism that plays a key role in mitochondria quality control and metabolic rewiring upon stress [10–12], which can reduce ROS levels. Functional loss of autophagy causes accumulation of defective mitochondria along with other autophagy substrates and can lead to oxidative stress, cell death and senescence [13–18]. Autophagy dysfunction is associated with neurological disorders such as Alzheimer's disease, Parkinson's disease, Huntington's disease and amyotrophic lateral sclerosis [19–23]. Neuron-specific ablation of essential autophagy genes *Atg5* or *Atg7* in mice results in motor and behavioral deficits, neurodegeneration, and greatly reduced survival [24,25].

Besides suppressing neurodegeneration, autophagy also has a role in promoting breast cancer [26,27]. Approximately 5–10% of breast cancers are familial, resulting from inherited mutations in tumor suppressor genes such as *BRCA1*, *BRCA2* and *PALB2* [28]. These genes encode large proteins that function in a BRCA1-PALB2-BRCA2 axis that is crucial for homologous recombination (HR)-mediated repair of DNA double strand breaks (DSBs) [29,30]. Moreover, we have shown that PALB2 promotes the accumulation and function of NRF2, a master antioxidant transcription factor, thereby promoting antioxidant response and reducing

ROS levels [31]. Interestingly, impaired autophagy, due to monoallelic loss of the essential autophagy gene *Becn1*, delayed and reduced *Palb2*-associated mammary tumorigenesis, suggesting that autophagy facilitates mammary tumor development following the loss of PALB2 [27].

In the current study, we originally aimed to co-delete *Palb2* and *Atg7* in the mammary gland using Cre recombinase driven by the promoter of whey acidic protein (*Wap-Cre*), to further study the role of autophagy in PALB2-associated breast cancer. Unexpectedly, *Wap-Cre* caused efficient deletion of the genes in the brain, and co-deletion of the two genes led to accelerated and more severe neurodegeneration compared with *Atg7* deletion alone. This was further supported by findings from mice with whole-body deletion of *Palb2* and *Atg7* driven by *Ubc-Cre-ERT2*. Detailed analyses of the brain revealed a novel role of PALB2 in regulating mitochondrial homeostasis and that the double deletion brain sustained more oxidative stress, rather than DNA damage, than either single deletion brain. Moreover, p53 was found to be induced by both *Palb2* and *Atg7* deletions, and further deletion of *Trp53* partially rescued the neurodegenerative phenotype of *Palb2/Atg7* double deletion mice. Our studies uncover a novel function of PALB2 and underscore the important roles and complex interplay of oxidative stress, autophagy, mitochondria and p53 in brain neurodegeneration.

## Results

### Survival, tumor development and motor behavioral deficits of mice with individual and combined ablation of *Palb2* and *Atg7*

As heterozygous disruption of *Becn1* delays *Palb2*-associated mammary tumor development [27], we sought to further examine the role of autophagy in mammary tumor development following PALB2 loss. *Palb2$^{f/f}$;Wap-Cre* (*Palb2*-CKO) mice were crossed with *Atg7*-floxed mice to generate mice with single and combined knockout of the two genes in the mammary gland. *Palb2*-CKO females showed a median overall survival ($T_{50}$) of 649 days (Fig 1A), and most of these mice developed spontaneous tumors (Fig 1B). Notably, only 6/19 (32%) of *Palb2*-CKO mice developed mammary tumors (Fig 1C), likely due to a mostly C57BL/6 genetic background, which is known to be more resistant to mammary tumorigenesis. *Atg7$^{f/f}$;Wap-Cre* (*Atg7*-CKO) mice had slightly shorter overall survival ($T_{50}$ = 604 days) (Fig 1A); 1 of the 22 mice (4.5%) developed mammary tumor and 9 (45.5%) developed other tumors (Fig 1C).

Unexpectedly, *Wap-Cre* driven *Atg7*-CKO mice showed progressive motor deficits, manifesting primarily as gait ataxia suggestive of cerebellar involvement. Note that although the *Wap-cre* used here was designed to express Cre specifically in the mammary gland, Cre activity has been reported in the brain [32]. Gait analysis at 6 weeks of age revealed an ataxic walking pattern of *Atg7*-CKO mice (Fig 1D). Compared with *Wap-Cre* (control) mice, mean stride length of *Atg7*-CKO mice was significantly decreased (Fig 1E), and both mean base width and overlap were significantly increased (Fig 1F and 1G). Beam balance test also showed gradual impairment of motor coordination in *Atg7*-CKO mice (Fig 1H and 1I). Additionally, these mice showed limb clasping reflexes when suspended by the tails (S1 Fig). These motor behavioral phenotypes are consistent with previous reports that loss of *Atg7* in the brain leads to neurodegeneration [17,33,34]. The mice were able to survive with motor deficits until approximately 400 days of age and then started to die. Eventually, 12/22 (54.5%) of the mice succumbed to neurodegeneration-associated death (Fig 1J), defined as death with severe motor deficits and without tumor or other overt pathological conditions seen upon dissection.

Remarkably, *Palb2$^{f/f}$;Atg7$^{f/f}$;Wap-Cre* (*Palb2;Atg7*-CKO) mice exhibited greatly reduced overall survival ($T_{50}$ = 280 days) (Fig 1A), even though their tumor development was also greatly reduced (Fig 1B). These double CKO mice showed more severe motor deficits,

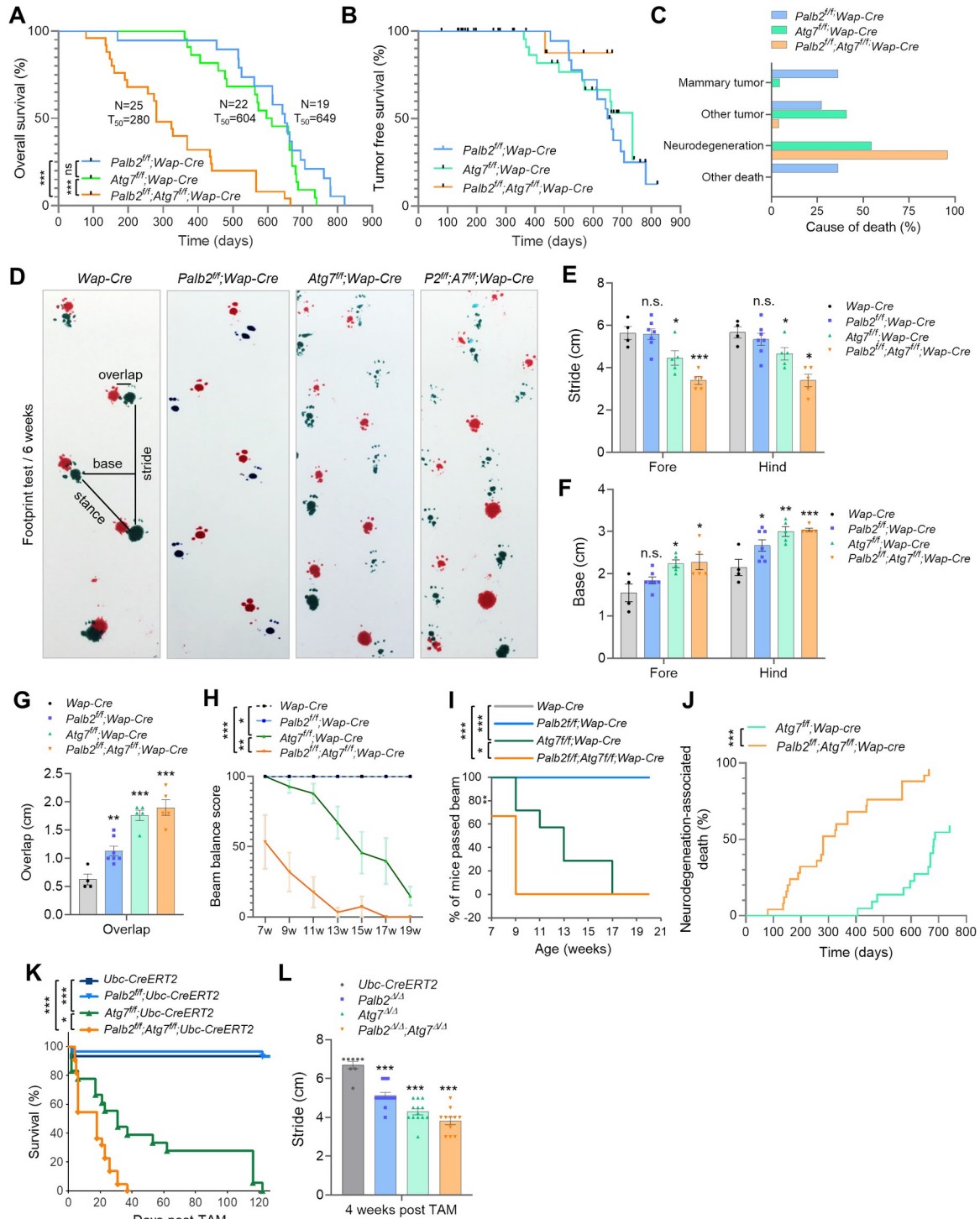

**Fig 1. Survival curves and motor deficits of *Palb2* and *Atg7* deletion mice.** (A-C) Overall survival (A), tumor-free survival (B) and observed causes of death (C) of *Wap-Cre* driven CKO mice. Black ticks in B indicate censoring events (mice that died without tumor). (D-G) Footprint test of the CKO mice at 6 weeks of age. Representative footprints and variables measured are shown in D, and results of stride, base and overlap measurements are shown in E, F and G, respectively. n = 4–7 mice per genotype. (H and I) Beam balance test of the CKO mice. Beam balance scores are shown in H and the percentage of mice that passed the beam in I. n = 4–7 mice per genotype. (J) Neurodegeneration-associated death in the CKO cohorts. (K-L) Survival curves (K) and stride length (L) of mice with *Ubc-Cre-ERT2* driven, whole body knockout (WBKO) of *Palb2* and/or *Atg7*. n = 8–13 mice per genotype.

especially in terms of stride length and balance (Fig 1E, 1H and 1I), and died from motor deficits much earlier than *Atg7*-CKO mice (Fig 1J). Similar to *Atg7*-CKO mice, these double CKO mice also showed limb clasping reflexes when suspended by the tails (S1 Fig). Only 1 out of 25 (4%) of these mice developed cancer, and all the rest died from severe motor deficits. The more severe motor deficits of the double CKO mice and the striking acceleration of death associated with neurodegeneration in these mice suggest that the two genes may synergistically suppress neurodegeneration. Although *Palb2*-CKO mice did not show any overt behavioral abnormalities, footprint tests showed increased hind base width and overlap (Fig 1F and 1G), suggestive of mild motor dysfunction.

We also crossed *Atg7*- and *Palb2*-floxed mice with mice carrying a *Ubc-Cre-ERT2* allele, which allow for tamoxifen (TAM)-induced whole-body knockout (WBKO) of floxed genes [35]. One week after (5 daily) TAM injections, efficient deletions of both *Palb2* and *Atg7* were observed in brain tissues (S2A Fig). The resulting *Palb2*-WBKO ($Palb2^{\Delta/\Delta}$) mice appeared normal and survived for at least a year. In contrast, a small number of $Atg7^{\Delta/\Delta}$ mice died from infections within the first week, the rest soon manifested motor deficits that worsened progressively over time, and the mice died within 120 days (Fig 1K), as we have shown recently [17,36]. Notably, $Palb2^{\Delta/\Delta};Atg7^{\Delta/\Delta}$ mice had even shorter survival compared with $Atg7^{\Delta/\Delta}$ mice ($T_{50}$ = 16 vs 30 days, p<0.05). Gait analysis at 4 weeks post TAM treatment showed that both $Palb2^{\Delta/\Delta}$ mice and $Atg7^{\Delta/\Delta}$ mice had significantly shortened hind stride length, and the length was even shorter in $Palb2^{\Delta/\Delta};Atg7^{\Delta/\Delta}$ mice (Fig 1L). These double knockout mice also died primarily from severe motor deficits, except those that died shortly after injection due to infections. Thus, loss of PALB2 in the brain leads to mild motor impairment but greatly exacerbates the neurological phenotype induced by *Atg7* deficiency.

## Loss of PALB2 exacerbates neurodegeneration induced by ATG7 loss

To better understand the cause of the neurological phenotype in our models, we assessed the "leaky" expression of Cre in multiple tissues. Indeed, deletion of *Palb2* was observed in the brains of *Palb2*-CKO and *Palb2;Atg7*-CKO mice (S2B and S2C Fig). Additionally, we analyzed the protein level of p62/SQSTM1, an autophagy adaptor protein that accumulates upon autophagy inhibition [37]. Western blotting showed strong p62 accumulation in midbrain, cerebellum and cerebral cortex in *Atg7*-CKO and *Palb2;Atg7*-CKO mice (S2D Fig). Immunofluorescence (IF) further revealed aberrant accumulation of p62 throughout the brain, with intense staining signals in midbrains and pons, and scattered signals in cerebral cortex and cerebellum (Figs 2A and 2B, and S3A and S3B). Significantly stronger p62 staining was found at 10 weeks than 6 weeks, and double CKO mice showed more p62 accumulation in midbrain and pons than did in *Atg7*-CKO mice at both ages. Further increased p62 accumulation was detected by immunohistochemistry (IHC) at 40 weeks (S4A Fig), indicative of a progressive accumulation of protein aggregates.

We also stained for tyrosine hydroxylase (TH), a marker of dopaminergic neurons in the substantia nigra that are degenerated in Parkinson's disease. No significant difference in TH staining was observed between control and any of the CKO mice at either 6 or 10 weeks of age (Fig 2C), indicating that the dopaminergic neurons were likely intact by 10 weeks of age, consistent with a previous report (Friedman et al., 2012). However, p62 accumulation was observed in TH+ neurons in both *Atg7*-CKO and double CKO mice at 6 weeks, and a further accumulation was seen at 10 weeks. Moreover, 10 weeks old double CKO mice had more TH +;p62+ neurons than *Atg7*-CKO mice, again suggesting that loss of PALB2 exacerbated p62 accumulation induced by loss of ATG7/autophagy (Fig 2C and 2D). Moreover, we observed significantly reduced TH+ neurons in the substantia nigra of double CKO mice than in *Atg7*-

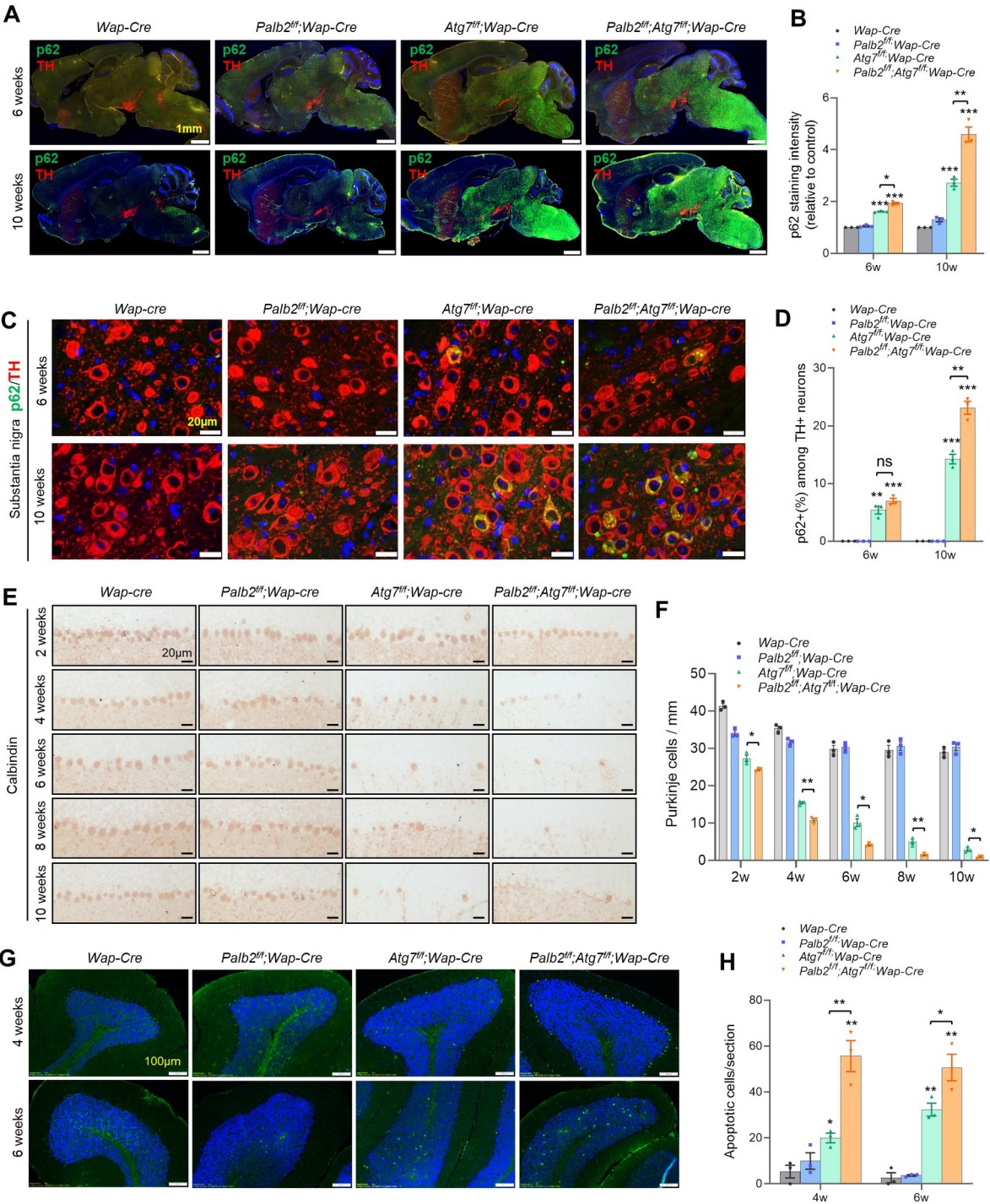

**Fig 2. Neuronal degeneration in *Palb2* and *Atg7* CKO mice.** (A-D) Representative immunofluorescence staining of p62/SQSTM1 and TH and quantification of their staining intensity in the whole midsagittal section (A and B) or substantia nigra (C and D) of the brain at 6 and 10 weeks of age. Nuclei were stained with DAPI. Images are representatives of two independent experiments. Scale bar = 1 mm in A and 20 μm in B as indicated. n = 3 mice per genotype. (E and F) Representative IHC staining of cerebellum with Purkinje cell marker Calbindin (E) and quantification of Purkinje cells (F) at 2, 4, 6, 8 and 10 weeks of age. n = 3 mice per genotype. Scale bar = 20 μm. (G and H) Representative TUNEL images (G) and quantification of positive cells (H) in the brains of CKO mice at 4 and 6 weeks of age. Scale bar = 100 μm. n = 3 mice per genotype.

CKO mice at 40 weeks (S4B Fig), suggesting that *Atg7* loss induced degeneration of dopaminergic neurons were also accelerated by *Palb2* loss.

It is well known that *Atg7* deletion in the brain leads to loss of Purkinje cells in the cerebellum [24,34], which are the primary controller of motor coordination and balance, and the motor deficits observed in the present study are consistent with degeneration of these neurons. Indeed, IHC for Calbindin, a marker for Purkinje cells, demonstrated progressive loss of these cells in *Atg7*-CKO mice between 2 and 10 weeks of age (Fig 2E and 2F). *Palb2*-CKO mice showed only a modest reduction in the number of these cells at 2 weeks but no further reduction later, while the double CKO mice showed more pronounced and significantly accelerated Purkinje cell loss compared with *Atg7*-CKO mice (Fig 2F). Apoptotic cells were detected in both Purkinje cell layer and granule cell layer of both *Atg7*-CKO and *Palb2;Atg7*-CKO mice, with the double CKO mice showing more apoptotic Purkinje cells than *Atg7*-CKO mice at 4 weeks of age (Fig 2G). At 6 weeks, while *Atg7*-CKO mice still had some apoptotic Purkinje cells, these cells were hardly detectable in *Palb2;Atg7*-CKO mice. At this age, the double CKO mice also had significantly more apoptotic cells in the granule cell layer than did *Atg7*-CKO mice (Fig 2G and 2H). Collectively, the above observations indicate that *Wap-Cre* driven deletion of *Atg7* induces neurodegeneration throughout the brain, with different neurons being affected differently in terms of timing and severity, and loss of *Palb2* exacerbates and accelerates the neurodegeneration induced by *Atg7* deletion. Since Purkinje cell is instantly recognizable and its loss is the most obvious histopathological change which also happens early and correlates with motor deficits caused by *Atg7* loss, we mainly focus on these cells in the following studies.

## DNA damage and oxidative stress in *Palb2* and *Atg7* deleted brains

To begin to understand the cause of the neurodegenerative phenotype of our mice and considering the known roles of PALB2 and autophagy in DNA repair and redox homeostasis, we assessed the levels of γH2AX, a marker for DSBs, 8-oxo-deoxyguanidine (8-oxo-dG), a marker for oxidative DNA damage, and 4-HNE, a product of lipid peroxidation, in cerebellum and midbrain of *Palb2* and *Atg7* CKO mice by IHC.

In cerebellum of both control and the CKO mice, positive γH2AX signals were only observed in Purkinje cells (Figs 3A and S5A and S5). Unlike the typical DNA damage-induced γH2AX nuclear foci, γH2AX in Purkinje cells was observed mostly in a single large, round structure in the nucleus, whose nature remains to be determined. At 6 weeks of age, about 16.6% of Purkinje cells in *Wap-Cre* control mice and 64.2% of Purkinje cells in *Palb2*-CKO mice showed positive γH2AX staining in the nucleus (Fig 3A and 3B). In *Atg7*-CKO mice and double CKO mice, most of the Purkinje cells had been lost, and the staining in remaining Purkinje cells cannot be interpreted with confidence due to cell shrinkage and possible ongoing apoptosis. In the midbrain, positive γH2AX signals were only observed in neurons and not in glial cells. Very few neurons in the control mice showed clear γH2AX foci and the signals were also weak, whereas most of neurons in *Palb2*-CKO, *Atg7*-CKO and *Palb2;Atg7* double CKO mice were positive and the signals were also much stronger (Fig 3A). Notably, *Palb2* single CKO neurons showed significantly more (but smaller) foci per cell, whereas *Atg7* single CKO neurons showed 2–3 large foci per cell. Compared to *Palb2* single CKO neurons, *Palb2;Atg7* double CKO neurons showed stronger staining signal and larger foci although fewer in number; compared to *Atg7* single CKO neurons, double CKO neurons showed both stronger staining signal and also more foci (Fig 3A and 3B).

γH2AX was also analyzed in the brains of the afore-described WBKO (Δ/Δ) mice. As expected, increased γH2AX signal was found in Purkinje cells of *Palb2*$^{Δ/Δ}$ mice at 4 weeks after

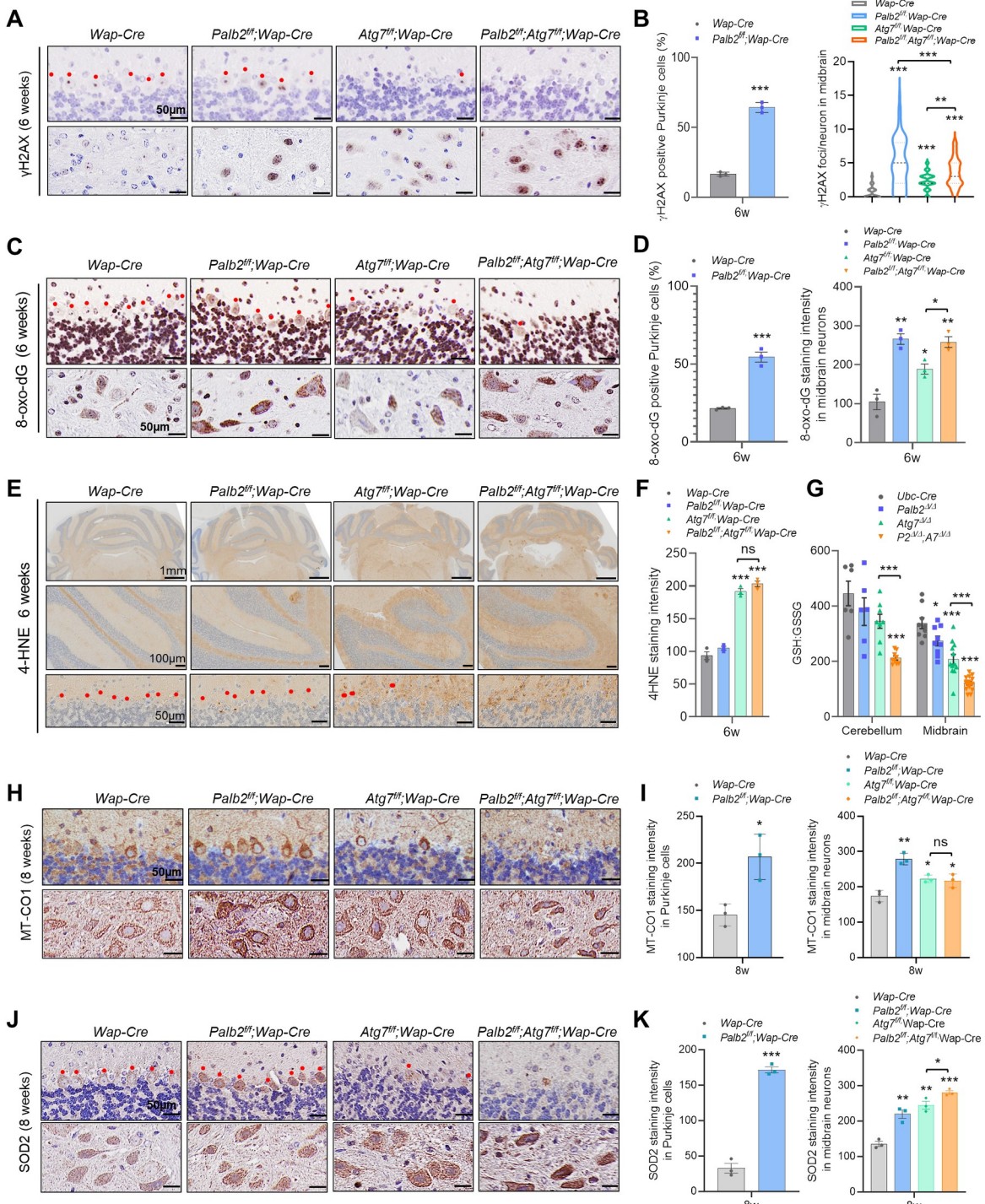

**Fig 3. Increased DNA damage and oxidative stress and mitochondrial markers in Purkinje cells of *Palb2* and *Atg7* deletion mice.**
(A-D) Representative IHC images and quantification of DNA DSB marker γH2AX (A and B) and oxidative DNA damage marker 8-oxo-dG (C and D) in cerebellum/Purkinje cells (upper images) and midbrain neurons (lower images) of the *Wap-Cre* driven CKO mice at 6 weeks. n = 3 mice per genotype. (E and F) Representative IHC images and quantification of lipid peroxidation marker 4-HNE in the cerebellum of the CKO mice at 6 weeks of age. Scale bar = 50 μM. n = 3 mice per genotype. (G) GSH:GSSG ratios in the cerebellum and midbrain of *Ubc-Cre-ERT2* driven WBKO mice. GSH and GSSG were measured by LC–MS/MS at 4 weeks after tamoxifen injection. n = 6–9 for cerebellum and n = 9–15 for midbrain. (H-K) Representative IHC images and quantification of MT-CO1 (H and I) and SOD2 (J and K) in cerebellum/Purkinje cells (upper images) and midbrain neurons (lower images) of CKO mice at 8 weeks. Red dots indicate Purkinje cells. Scale bar = 50 μm. n = 3 mice per genotype.

tamoxifen injection (S6A Fig). At this time, Purkinje cells had been mostly lost in $Atg7^{\Delta/\Delta}$ and $Palb2^{\Delta/\Delta};Atg7^{\Delta/\Delta}$ mice; however, increased γH2AX was detected in a subset of granule cells in these mice, suggesting that ATG7 plays a role, directly or indirectly, in DNA damage repair, consistent with our recent findings [17]. Moreover, we subjected the mice to 10 Gy of whole-body gamma radiation. As expected, $Palb2^{\Delta/\Delta}$ mice showed substantially reduced survival compared with either control or $Atg7^{\Delta/\Delta}$ mice (S6B Fig). Notably, however, $Palb2^{\Delta/\Delta};Atg7^{\Delta/\Delta}$ mice exhibited similar survival to that of $Palb2^{\Delta/\Delta}$ mice post radiation, suggesting that the DNA repair deficit of $Palb2^{\Delta/\Delta};Atg7^{\Delta/\Delta}$ mice stems primarily from PALB2 deficiency.

In the cerebellum, 8-oxo-dG staining signals were observed in both Purkinje and granular cells, but any clear difference between control and CKO mice was mainly found in Purkinje cells (S5C and S5D Fig). At 2 weeks of age, very few Purkinje cells in the control mice showed clear positive 8-oxo-dG staining and the signals were also weak, whereas more cells in *Palb2*-CKO and *Atg7*-CKO mice were positive and the signals were slightly stronger. Notably, *Palb2; Atg7*-CKO mice had even more 8-oxo-dG positive Purkinje cells at this time point, and the signals were also substantially stronger. At 4 weeks, the staining in *Palb2*-CKO mice was much stronger than that at 2 weeks and also much stronger than that in either control or *Atg7*-CKO mice, although *Atg7*-CKO mice showed slightly higher staining intensity than the control. At 6 weeks of age, about 22% of Purkinje cells in the control mice showed clear positive 8-oxo-dG staining and the signals were also weak, whereas 54% cells in *Palb2*-CKO mice were positive and the signals were much stronger (Fig 3C and 3D). In midbrain of 6 weeks mice, since most of neurons showed positive 8-oxo-dG staining, we only measured staining intensity. Compared with control mice, much stronger 8-oxo-dG staining was found in both *Palb2*-CKO mice and double CKO mice, and *Atg7*-CKO mice showed moderately higher staining intensity (Fig 3C and 3D). These results suggest that PALB2 is critical for preventing DNA oxidation in both Purkinje cells and midbrain neurons, whereas ATG7 plays a less prominent role. On the other hand, 4-HNE, a lipid peroxidation by-product, was evidently more intense in *Atg7*-CKO and *Palb2;Atg7*-CKO brains (Fig 3E and 3F). Thus, ATG7 and autophagy play a more important role in suppressing lipid peroxidation than does PALB2.

Finally, we also determined the ratio of reduced to oxidized glutathione (GSH:GSSG). Reduced glutathione (GSH) is known to play a pivotal role in the cellular oxidant defense system and is indispensable for preventing lipid peroxidation by free radicals and oxygen [38], and decreased GSH:GSSG ratio is a key indicator of decreased ability to reduce cytotoxic lipid peroxides [39]. Cerebellum and midbrain were collected from the WBKO mice and tissue metabolites were measured by liquid chromatography-mass spectrometry (LC-MS). As shown in Fig 3G, in the brain of the WBKO mice, GSH:GSSG ratio was modestly (though not statistically significantly) decreased in the cerebellum of both $Palb2^{\Delta/\Delta}$ and $Atg7^{\Delta/\Delta}$ mice, and the ratio was further and statistically significantly decreased in the cerebellum of $Palb2^{\Delta/\Delta};Atg7^{\Delta/\Delta}$ mice. In the midbrain, the same trends were observed, and all the differences were larger and statistically significant. These results further support the increase in lipid peroxidation in the cerebellum and midbrain induced by combined deletion of *Palb2* and *Atg7*. Taken together, our results reveal both similar and distinct impacts of PALB2 and ATG7 loss on DNA damage and oxidative stress in the brain.

## Requirement of PALB2 for mitochondrial homeostasis in the brain

Given the key role of autophagy in mitochondrial quality control and mitochondrial dysfunction in neurodegenerative diseases, we analyzed two mitochondrial proteins by IHC. First, we examined Cytochrome c oxidase subunit I (MT-CO1), a subunit of respiratory Complex IV, which is located within the mitochondrial inner membrane and is the final enzyme of the

electron transport chain of mitochondrial oxidative phosphorylation [40]. At 2 weeks of age, both *Palb2*-CKO and *Atg7*-CKO mice appeared to have slightly higher levels of MT-CO1 than control; at 4 weeks, MT-CO1 signals in *Palb2*-CKO Purkinje cells were much stronger than control, while signals in *Atg7*-CKO and *Palb2;Atg7* double CKO cells were hard to interpret due to cell shrinkage (S5E and S5F Fig). At 8 weeks of age, both Purkinje cells and midbrain neurons had much higher staining intensity of MT-CO1 in *Palb2*-CKO mice than in control; in *Atg7*-CKO and *Palb2;Atg7* double CKO midbrain neurons, MT-CO1 was moderately higher than that in control (Fig 3H and 3I). Additionally, increased MT-CO1 was also found in Purkinje cells of *Palb2*$^{\Delta/\Delta}$ mice (S6A Fig). Next, we stained for SOD2 (MnSOD), a superoxide dismutase that is localized in the mitochondrial intermembrane space and converts superoxide anion resulting from leaked electrons from the electron transport chain to hydrogen peroxide ($H_2O_2$) [41]. At 8 weeks, both Purkinje cells and midbrain neurons in *Palb2*-CKO mice showed much stronger SOD2 staining than controls, signals in *Atg7*-CKO midbrain neurons were even stronger than in *Palb2*-CKO mice, while neurons in the double CKO midbrain showed strongest staining (Fig 3J and 3K).

## Regulation of mitochondrial gene expression by PALB2 in human medulloblastoma cells

To confirm the role of PALB2 in regulating mitochondria, we conducted RNA-seq analysis following siRNA-mediated knockdown of *PALB2*. Six different *PALB2* siRNAs (along with five controls) were individually transfected into DAOY medulloblastoma cells, which have a cerebellar origin [42], to deplete the protein (Fig 4A). A total of 540 genes were significantly up- or down-regulated in *PALB2* knockdown cells (S1 File). Top 50 upregulated and top 50 downregulated differentially expressed genes (DEGs) affected by *PALB2* knockdown are shown in Fig 4B. Gene set enrichment analysis (GSEA) revealed highly enriched DEGs in oxidative phosphorylation (Fig 4C and 4D), mitochondrial gene expression (Fig 4E and 4F), mitochondrial translation, respiratory chain complex assembly, mitochondrial electron transport chain, ATP synthesis coupled electron transport and mitochondrial complex I (NADH:ubiquinone oxidoreductase) (S1 File), again implicating PALB2 in mitochondrial regulation. Quantitative reverse transcription PCR (qRT-PCR) further validated the RNA-seq results and confirmed the upregulation of *MT-CO1*, *MT-CO2* and *COX4i1*, which encodes Cytochrome c oxidase subunit 4 isoform 1, another subunit of respiratory Complex IV, in *PALB2* knockdown cells. Furthermore, the expression of *PGC1A*, encoding peroxisome proliferator-activated receptor gamma coactivator 1-alpha, a transcriptional coactivator and central inducer of mitochondrial biogenesis [43,44], was also increased in *PALB2* knockdown cells, supporting a role for PALB2 in mitochondrial biogenesis. Next, we analyzed mitochondria using MitoTracker Green and MitoTracker Red, which measure mitochondrial mass and mitochondrial membrane potential, respectively. Both fluorescence microscopic visualization and flow cytometry analysis showed significantly increased mitochondrial mass and membrane potential in *PALB2* knockdown cells (Fig 4H–4J).

## Mitochondrial abnormality, oxidative stress, and cell death upon combined loss of PALB2 and ATG7 in human medulloblastoma cells

To further assess the role of PALB2 in mitochondrial biogenesis/function and its functional relationship with ATG7 in human cells, we used CRISPR/Cas9 to knock out *PALB2* and *ATG7* in the DAOY medulloblastoma cells. Clones were screened using western blotting (S7A Fig). *PALB2;ATG7* double knockout (DKO) clones were then generated by further knocking out *ATG7* in a sequence validated and functionally characterized *PALB2*-KO clone. Consistent

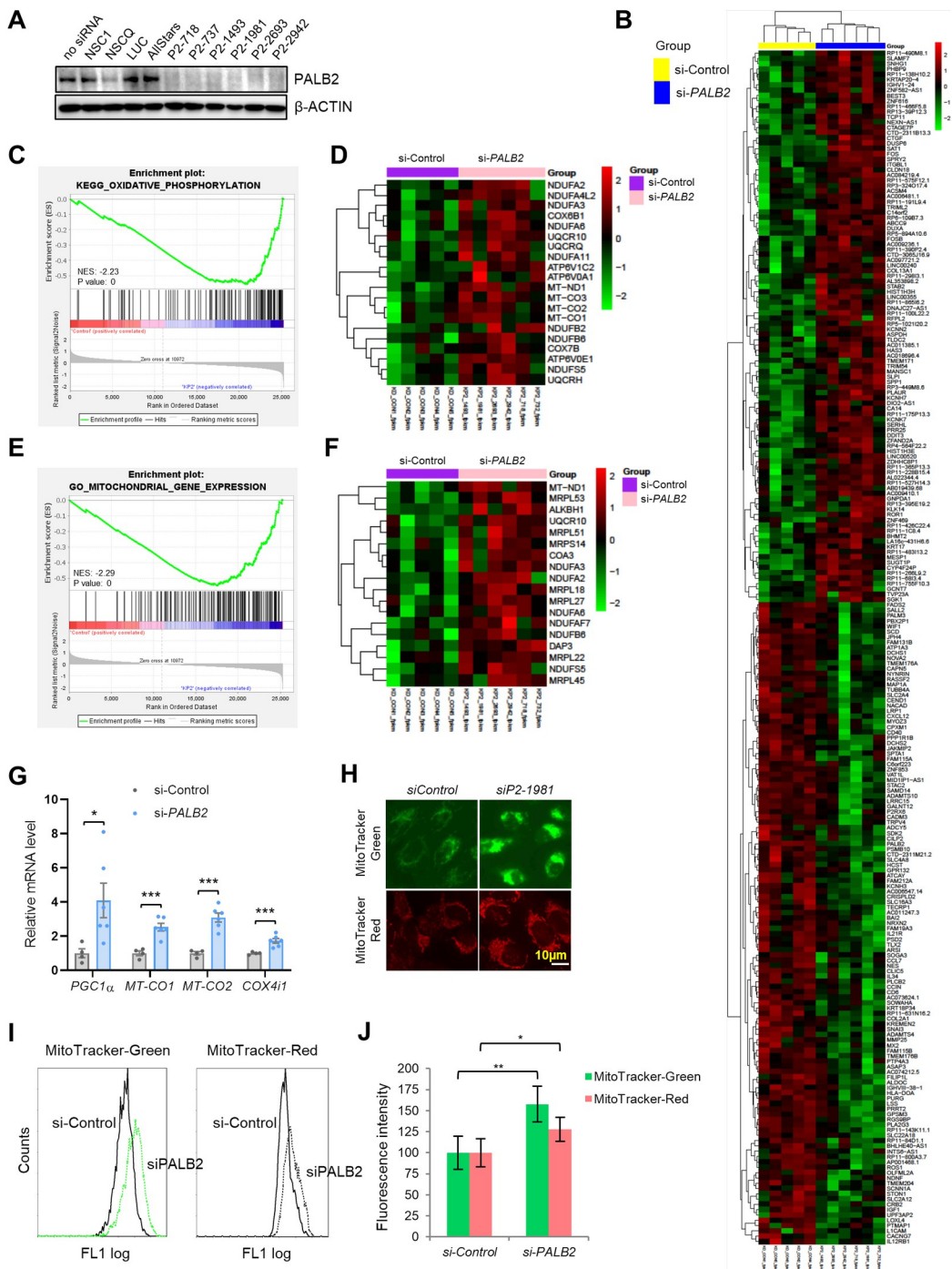

**Fig 4. Impact of PLAB2 loss on mitochondrial gene expression and function in human medulloblastoma cells.** (A) Western blot analysis of PALB2 levels in DAOY cells treated with transfection reagent alone, 4 different control siRNAs and 6 different PALB2 siRNAs. β-ACTIN was used as loading control. (B) Heat map showing top 50 upregulated and top 50 downregulated genes from two-way hierarchical clustering analysis of the 11 samples (5 control and 6 si*PALB2*). (C and D) GSEA enrichment plot (C) and heat map (D) of the expression of genes in the "oxidative phosphorylation" gene set in control and *PALB2* knockdown cells. (E and F) GSEA enrichment plot (E) and heat map (F) of the expression of genes in the "mitochondrial gene expression" gene set in control and *PALB2* knockdown cells. (G) qRT–PCR analysis of the expression of *PGC1α*, *MT-CO1*, *MT-CO2* and *COX4i1* in control and PALB2 knockdown cells. Results are averages of the 5 control and 6 *PALB2* knockdown cells from three independent experiments, each performed in two technical replicates. (H) Representative fluorescence images of MitoTracker Green and MitoTracker Red from control and *PALB2* knockdown cells. Scale bar = 10 μm. (I and J) Representative flow cytometry overlay plot (I) and quantified fluorescence intensity (J) of

MitoTraker Green and MitoTracker Red signals in the above control and PALB2 knockdown cells. n = 3 independent experiments.

with the essential role of ATG7 in LC3 lipidation and autophagosome formation [45], *ATG7*-KO and *PALB2;ATG7*-DKO cells showed reduced LC3-I to LC3-II conversion and increased p62 levels (Fig 5A). MT-CO1, MT-CO2 and SOD2 were all increased in *PALB2*-KO, *ATG7*-KO and *PALB2;ATG7*-DKO cells, implying increased mitochondrial mass upon loss of either PALB2 or ATG7. The growth rate of *ATG7*-KO cells was slightly slower than that of control cells, while *PALB2*-KO cells grew significantly slower, and a further decrease in growth rate was seen for *PALB2;ATG7*-DKO cells (Fig 5B). Both *PALB2*-KO and *ATG7*-KO cells showed moderate boundary shrinkage, and the DKO cells showed overt boundary shrinkage along with soma cavitation (Fig 5C). Spontaneous cell death, including both apoptosis and necrosis, was increased in *PALB2*-KO and *ATG7*-KO cells and further increased in the DKO cells (Fig 5D and 5E).

We also analyzed mitochondria using MitoTracker Green and MitoTracker Red in all the positive clones at early passage and quantified the signals by flow cytometry (Figs 5F and 5G, and S7A and S7B). Consistent with previous reports [36,46,47], *ATG7*-KO cells showed an increase in total mitochondrial mass. Notably, an even larger increase in mitochondrial mass was observed in *PALB2*-KO cells. Both *PALB2* and *ATG7* single KO cells showed modestly increased membrane potential compared with WT cells, whereas the DKO cells showed no change. The ratio of MitoTracker Red to Green signals (R/G) of *PALB2*-KO, *ATG7*-KO and the DKO cells was 85.09%, 92.82% and 88.76%, respectively, of wt level, indicative of impaired mitochondrial quality in all 3 cell types (Fig 5F and 5G). The increased mitochondrial mass in *PALB2*-KO cells corroborates with our RNA-seq results (Fig 4) and in vivo findings (Fig 3) and lends further support to the notion that loss of PALB2 leads to increased mitochondrial biogenesis, while the decreased R/G ratio in the cells suggests lower overall mitochondria quality or a defect in the clearance of damaged mitochondria. To evaluate if mitochondrial respiration was affected by *PALB2* and/or *ATG7* inactivation, oxygen consumption rate (OCR) was measured using the Seahorse assay. Basal mitochondrial respiration, ATP-linked respiration and maximal respiration were moderately decreased in both *PALB2*-KO and *ATG7*-KO cells, but dramatically decreased in *PALB2;ATG7*-DKO cells (Fig 5H and 5I). These data suggest that a strong defect in mitochondrial respiration may be a major contributor to the spontaneous death of DKO cells.

Mitochondria are both a major producer and a primary target of ROS [48]. Consistent with our in vivo observations, cellular ROS level measured by DCF, which mainly detects $H_2O_2$, was significantly higher in *PALB2*-KO cells and even higher in *PALB2;ATG7*-DKO cells; however, no change in DCF readout was observed in *ATG7*-KO cells (Fig 5J). In contrast, mitochondrial superoxide, as measured by MitoSOX, was increased in *ATG7*-KO cells but not in *PALB2*-KO or *PALB2;ATG7*-DKO cells (Fig 5K). The increased superoxide level in *ATG7*-KO cells is consistent with recent reports [49,50], and the lack of any change in DCF readings suggests that the overproduced superoxide in these cells was not converted to $H_2O_2$ in a timely fashion, or that any increase in $H_2O_2$ had been promptly disposed of or reacted with polyunsaturated fatty acids leading to lipid peroxidation. On the other hand, the increase in DCF signals in *PALB2*-KO cells without any concomitant increase in superoxide suggests that the higher ROS level may not originate from mitochondria, or the overproduced superoxide has been converted to $H_2O_2$ by SOD2. These data are largely consistent with our findings from the mouse brain and support the notion that a combination of abnormal mitochondria function and increased oxidative stress leads to apoptosis in Purkinje cells and certain other neurons, hence the motor deficits.

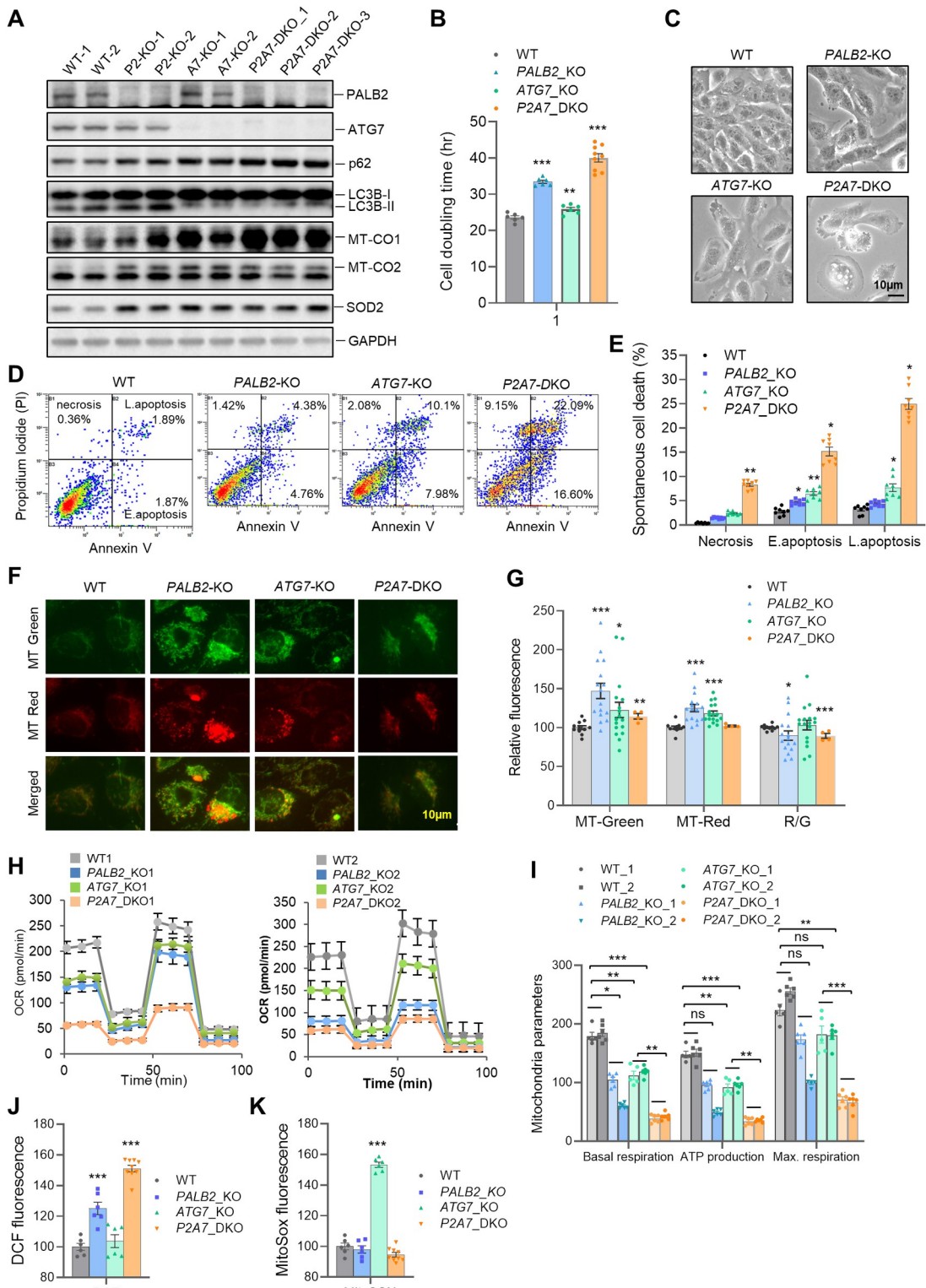

**Fig 5. Mitochondrial abnormality, oxidative stress, and apoptosis in *PALB2* and *ATG7* KO human medulloblastoma cells.**
(A) Western blot analysis of whole cell homogenates of WT and KO cells. All 9 cell lines were from single clones. The WT cell lines were false positive clones from the *PALB2* CRISPR/Cas9 KO procedure. (B) Doubling times of WT and KO cells of the 9 cell lines in A. n = 3 independent experiments. Cells of the same genotype were grouped together. (C) Representative bright field microscopy images of the cells. Scale bar = 10 μm. (D and E) Representative flow cytometry plots (D) and quantification (E) of

FITC Annexin V analysis of the cells. n = 3 independent experiments, each performed in technical duplicates. Cells of the same genotype were grouped together in E. (F and G) Mitochondrial analyses of WT and KO cells with MitoTracker-Green and MitoTracker-Red. Representative images of stained cells are shown in F. Flow cytometry-based quantification of the signals in all cell clones are shown in G. Avg. R/G, average ratios of Red vs Green signals (in percentage). Scale bar = 10 μm. (H) Oxygen consumption rate (OCR) in WT and KO cells as measured by the Seahorse assay. n = 5 technical replicates. (I) Bar graph of mitochondrial functional parameters as measured in panel H. (J and K) Levels of reactive oxygen species in WT and KO cells as measured by DCF (J) and MitoSOX (K) staining followed by flow cytometry-based detection and quantification.

## Loss of BRCA2 delays the onset of neurodegeneration induced by ATG7 deficiency

To assess whether an HR deficiency and increased DSB formation were a significant cause of the more severe neurodegeneration in *Palb2*;*Atg7*-CKO mice than in *Atg7*-CKO mice, we analyzed *Brca2^{f/f}*;*Atg7^{f/f}*;*Wap-cre* (*Brca2*;*Atg7*-CKO) mice. Interestingly, unlike *Palb2*;*Atg7*-CKO mice, *Brca2*;*Atg7*-CKO mice had similar overall survival and moderately reduced neurodegeneration-associated death compared with *Atg7*-CKO mice (Fig 6A–6C). Also unlike *Palb2*, co-deletion of *Brca2* with *Atg7* did not exacerbate the neurological phenotype of *Atg7*-CKO mice; instead, it significantly moderated Purkinje cell loss caused by *Atg7* deficiency from 2 to 10 weeks of age (Fig 6D). Both gait analysis and beam balance test also revealed a rescue of motor coordination and balance defects by co-deletion of *Brca2* with *Atg7* (Fig 6E and 6F).

To understand why ablation of *Brca2* produced a different effect from that of *Palb2* ablation, we compared γH2AX, 8-oxo-dG and MT-CO1 in the brains of the mice at 6 weeks of age. As shown in Fig 6G–6J, γH2AX positivity in Purkinje cells of *Brca2*-CKO mice was slightly lower than that in *Palb2*-CKO mice but still much higher than that in control mice; in contrast, their 8-oxo-dG and MT-CO1 staining signals were substantially weaker than those in *Palb2*-deleted Purkinje cells. The modestly higher γH2AX positivity in *Palb2*-deleted Purkinje cells compared with *Brca2*-deleted cells may be attributed to higher oxidative stress, which can cause DSBs. Compared with the remaining Purkinje cells in *Atg7*-CKO mice, the same cells in *Brca2*;*Atg7*-CKO mice showed similar γH2AX positivity. The staining patterns of both MT-CO1 and 8-oxo-dG in Purkinje cells of *Brca2*;*Atg7*-CKO mice were also similar to that in *Atg7*-CKO mice. Therefore, loss of DNA repair function of PALB2 is unlikely to be a significant cause of the more severe neurological phenotype of *Palb2*;*Atg7*-CKO mice, and that increased oxidative stress and/or mitochondrial dysfunction may underlie this exacerbated phenotype.

## Loss of p53 delays Purkinje cell death and prolongs the survival of *Palb2*; *Atg7*-CKO mice

Our recent study showed that ATG7 limits p53 activation and p53-induced neurodegeneration [17]. To determine whether p53 played a role in the shortened survival of *Palb2*;*Atg7*-CKO mice, we set up cohorts of these mice either without or with floxed *Trp53*. Compared with *Palb2*;*Atg7*-CKO mice, overall survival of *Palb2*;*Atg7*;*Trp53*-CKO mice was significantly prolonged (T$_{50}$ = 448 vs 280 days, p = 0.0165) (Fig 7A), and neurodegeneration-associated death was greatly reduced and delayed (Fig 7B). Moreover, inactivation of *Trp53* combined with prolonged survival allowed for efficient tumor development (Fig 7C). While 95% of *Palb2*;*Atg7*-CKO mice died from neurodegeneration, 45% of *Palb2*;*Atg7*;*Trp53*-CKO mice developed mammary tumors and 21% developed other tumors, with neurodegeneration causing the death of only 30% of the mice.

The above results suggest that p53 was activated upon loss of PALB2 and/or ATG7 in the brain which subsequently led to neuronal apoptosis. Indeed, IHC analysis revealed that at 2 weeks of age, p53 protein level was moderately increased in *Atg7*-CKO Purkinje cells in terms

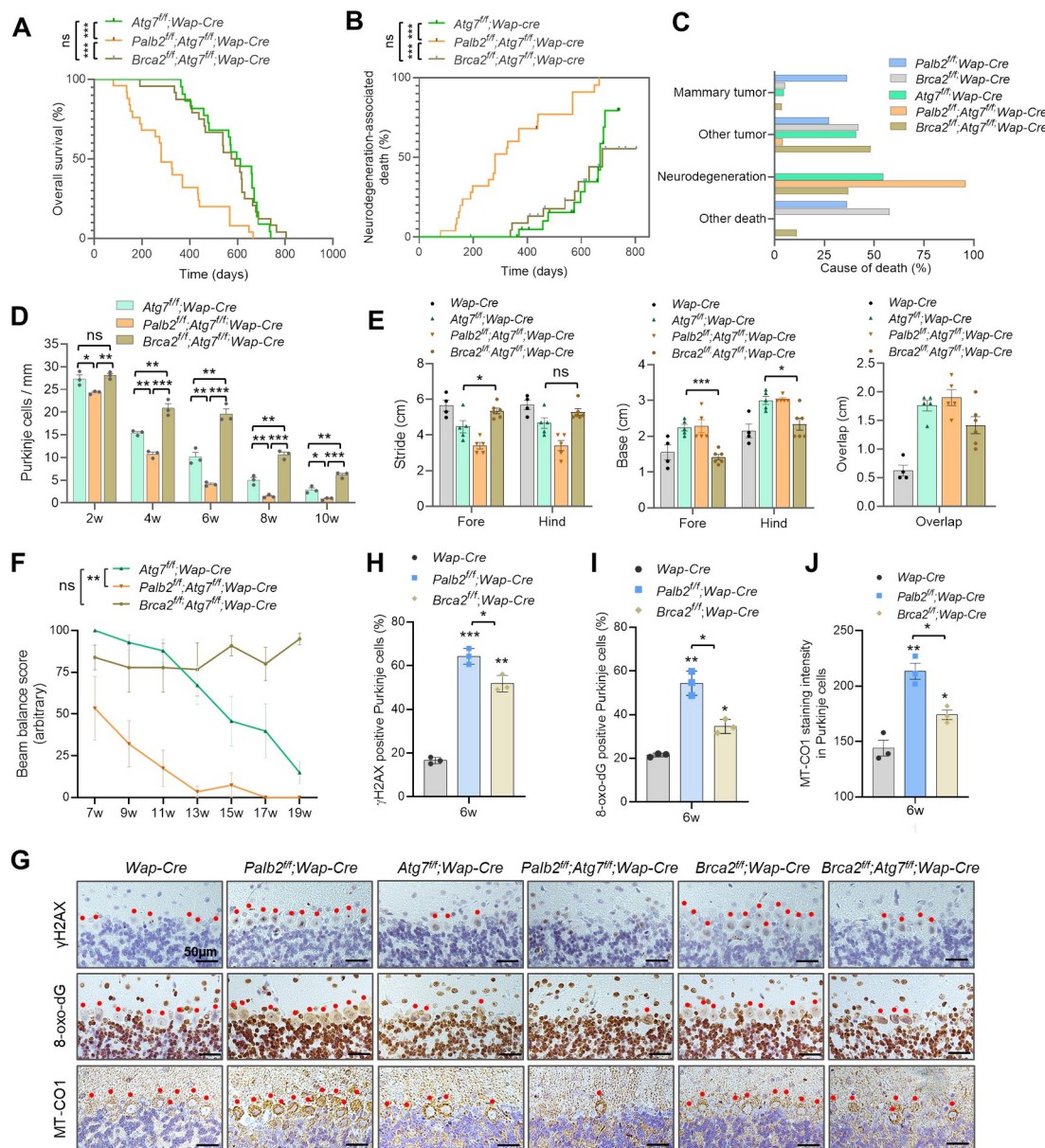

**Fig 6. Comparative analyses of the impacts of *Brca2* and *Palb2* deletions in wt and *Atg7*-CKO mice.** (A-C) Overall survival (A), neurodegeneration-associated death (B) and cause of death (C) of the CKO mice. *Atg7^{f/f};Wap-Cre*, n = 22; *Palb2^{f/f}; Atg7^{f/f}; Wap-Cre*, n = 25; *Brca2^{f/f}; Atg7^{f/f};Wap-Cre*, n = 24. (D) Purkinje cell numbers in CKO mice at 2 to 10 weeks of age. n = 3 mice per genotype per time point. (E) Footprint measurements of 6 weeks old CKO mice. n = 5–7 mice per genotype. (F) Beam balance scores of the CKO mice at different ages. (G) Representative images of γH2AX, 8-oxo-dG and MT-CO1 IHC in the cerebellum of 6 weeks old CKO mice. Red dots indicate Purkinje cells. Scale bar = 50 μm. (H-J) Quantification of γH2AX positivity (H), 8-oxo-dG positivity (I) and MT-CO1 staining intensity (J) in Purkinje cells (n = 3 mice per genotype).

of both staining intensity and number of positive cells, and even higher p53 accumulation was seen in Purkinje cells of *Palb2*-CKO and the double CKO mice (Fig 7D and 7E). The overall situation was similar at 4 weeks, except that some or most Purkinje cells in *Atg7*-KO mice and *Palb2*;*Atg7*-CKO mice had been lost.

Co-deletion of *Trp53* produced no significant effect on Purkinje cell numbers in control, *Palb2*-CKO or *Atg7*-CKO mice; however, the severe loss of Purkinje cells in *Palb2*;*Atg7*-CKO

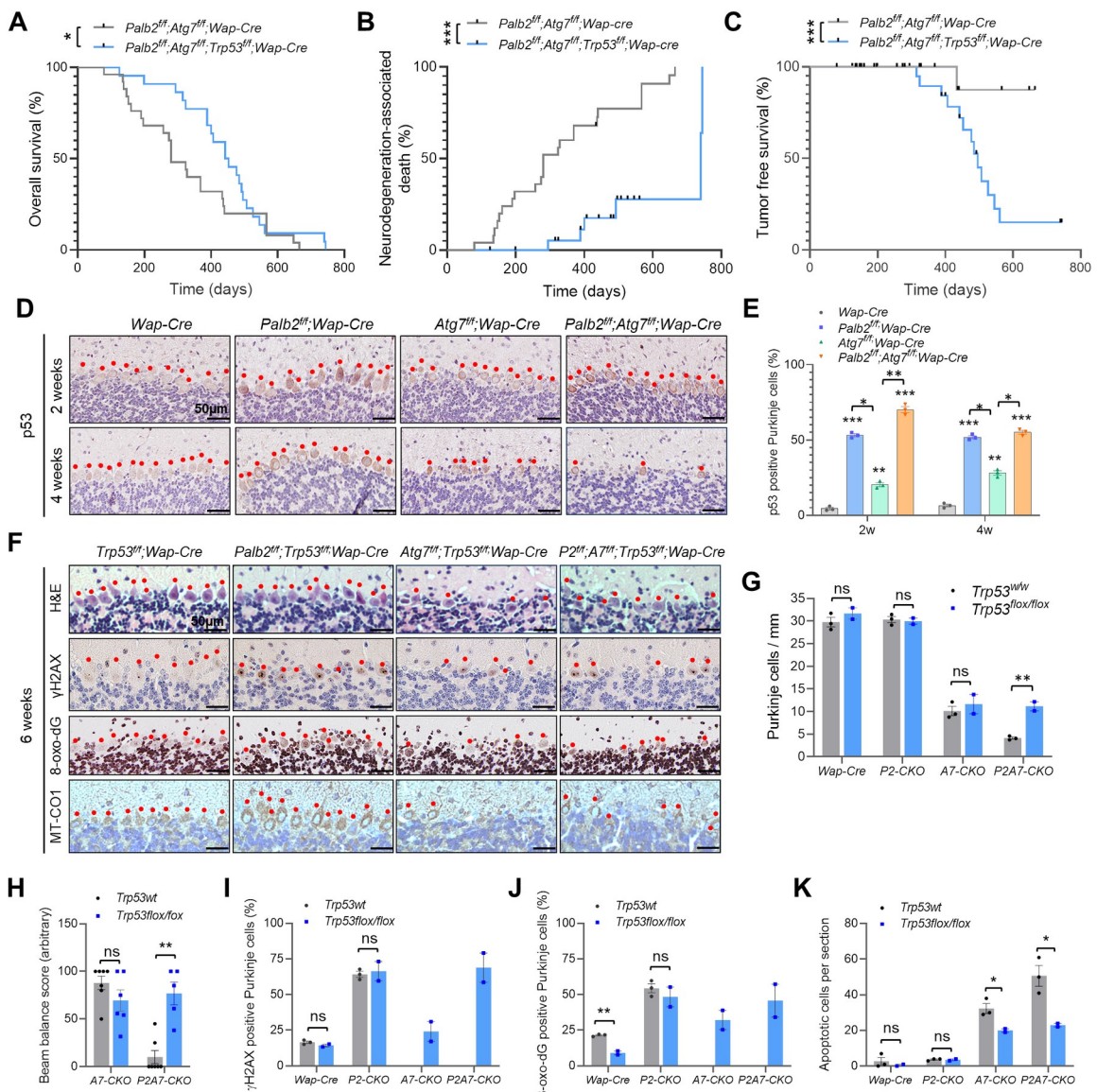

**Fig 7. Role of p53 in Purkinje cell death and the survival of *Palb2;Atg7*-CKO mice.** (A-C) Overall survival (A), neurodegeneration-associated death (B), and tumor free survival (C) of *Palb2;Atg7*-CKO and *Palb2;Atg7;Trp53*-CKO mice. Where applicable, black ticks indicate censoring events. *Palb2;Atg7*-CKO, n = 25; *Palb2;Atg7;Trp53*-CKO, n = 22. (D and E) Representative p53 IHC images (D) and quantification of p53 positive Purkinje cells (E) in the cerebellum of control and CKO mice at 2 and 4 weeks of age. Red dots indicate Purkinje cells. Scale bar = 50 μm. n = 3 mice per genotype per time point. (F) Representative H&E and IHC images of γH2AX, 8-oxo-dG and MT-CO1 in the cerebellum of the CKO mice at 6 weeks of age. Images are representatives from two independent experiments. Red dots indicate Purkinje cells. Scale bar = 50 μm. (G) Purkinje cell numbers in the CKO mice at 6 weeks of age, n = 2–3 mice per genotype. (H) Beam balance scores of the CKO mice at 11 weeks of age. n = 5–7 mice per genotype. (I and J) Quantification of γH2AX positive (I) and 8-oxo-dG positive (J) Purkinje cells at 6 weeks of age. n = 2–3 mice per genotype. (K) Number of apoptotic cells in the cerebellum of 6 weeks old control and CKO mice. n = 2–3 mice per genotype.

mice was significantly rescued (Fig 7F and 7G). Interestingly, when *Trp53* was co-deleted, Purkinje cell number in *Palb2;Atg7*-CKO mice was restored to a level similar to that in *Atg7*-CKO mice (Fig 7G), suggesting that the severe neurodegenerative phenotype in *Palb2;Atg7*-CKO mice is likely an *Atg7* null phenotype exacerbated by a further induction of p53 elicited by loss of PALB2. Consistent with the rescue of Purkinje cell number, motor function of *Palb2;Atg7;*

*Trp53*-CKO mice was substantially preserved, with the mice achieving similar beam balance scores as *Atg7*-CKO mice at 11 weeks of age (Fig 7H).

Deletion of *Trp53* alone (in *Wap-Cre* control mice) did not cause any significant effect on γH2AX staining in Purkinje cells but led to a substantial reduction in 8-oxo-dG positivity (Fig 7I and 7J), and the low basal level of apoptosis (detected in whole brain sagittal sections) was unaffected (Fig 7K). Co-ablation of *Trp53* with *Palb2* had no effect on any of these three parameters, whereas inactivation of *Trp53* in *Atg7*-CKO and *Palb2*;*Atg7*-CKO mice led to a decrease in all three parameters (Fig 7I–7K); however, the results of γH2AX and 8-oxo-dG need to be interpreted with caution because most of the Purkinje cells were lost in mice with wt p53. Taken together, these results demonstrate an important role for p53 in inducing neuronal death upon combined loss of PALB2 and ATG7/autophagy.

## Partial rescue of survival of *Palb2*$^{\Delta/\Delta}$;*Atg7*$^{\Delta/\Delta}$ mice by an ROS scavenger

Collectively, results presented above suggest that excessively high oxidative stress, rather than increased DNA damage, in the brain, particularly Purkinje cells, underlies the severe neurodegenerative phenotype of *Palb2*;*Atg7*-CKO and *Palb2*$^{\Delta/\Delta}$;*Atg7*$^{\Delta/\Delta}$ mice. To test this notion, we treated *Palb2*$^{\Delta/\Delta}$;*Atg7*$^{\Delta/\Delta}$ mice with N-acetylcysteine (NAC), an ROS scavenger, starting from the first day of 4 daily TAM injections, and monitored their survival. Compared with untreated mice, NAC-treated animals showed similar survival during the first 35 days or so and then survived significantly longer (Fig 8A and 8B). The divergence suggests that there likely were two different causes of death in the mice and that NAC protects against late death or the second cause, which was neurodegeneration. The early death appeared to be caused by tamoxifen toxicity or Streptococcus infection as we showed in our previous study [51].

To test the effect of NAC on DNA damage, oxidative stress, and neuronal survival, we separately treated another set of *Palb2*$^{\Delta/\Delta}$;*Atg7*$^{\Delta/\Delta}$ mice with NAC for 8 days, also starting from the first day of 4 daily TAM injections, which generated efficient deletion of the genes (Fig 8C). IHC results showed that by the end of the 8 days, Purkinje cells in NAC untreated mice were almost all lost, and this loss was strongly rescued by NAC treatment (Fig 8D and 8E). As few Purkinje cells remained in the control mice, comparison of DNA damage and oxidative stress was not feasible in these cells. However, in midbrain neurons, it was evident that NAC treatment significantly reduced 8-oxo-dG but not γH2AX staining (Fig 8D and 8E). These results provide further support that excessive oxidative stress, rather than increased DNA damage, caused by co-deletion of *Palb2* with *Atg7* underlies the severe neurodegenerative phenotype of *Palb2*;*Atg7*-CKO and *Palb2*$^{\Delta/\Delta}$;*Atg7*$^{\Delta/\Delta}$ mice.

## Discussion

Autophagy plays important roles in redox homeostasis, mitochondrial quality control, and genome maintenance [52]. It is well established that basal levels of autophagy protect against neurodegeneration, as has been shown by various mouse models with conditional deletion of *Atg7* [24,34,53,54]. In this study, our results from unintended *Wap-Cre* driven deletion of *Atg7* in the brain and intended *Ubc-Cre-ERT2* driven whole-body knockout of the gene in adult mice both showed that loss of ATG7 in the brain leads to apoptosis of neurons, especially Purkinje cells, motor deficits and shortened survival. Moreover, *Atg7* deletion in the brain led to increased mitochondrial mass, reduced GSH/GSSG ratio, increased DNA and lipid oxidation, as well as increased DNA damage. These results are largely consistent with published reports and further underscore the key roles of autophagy in the above-noted processes and neuronal health.

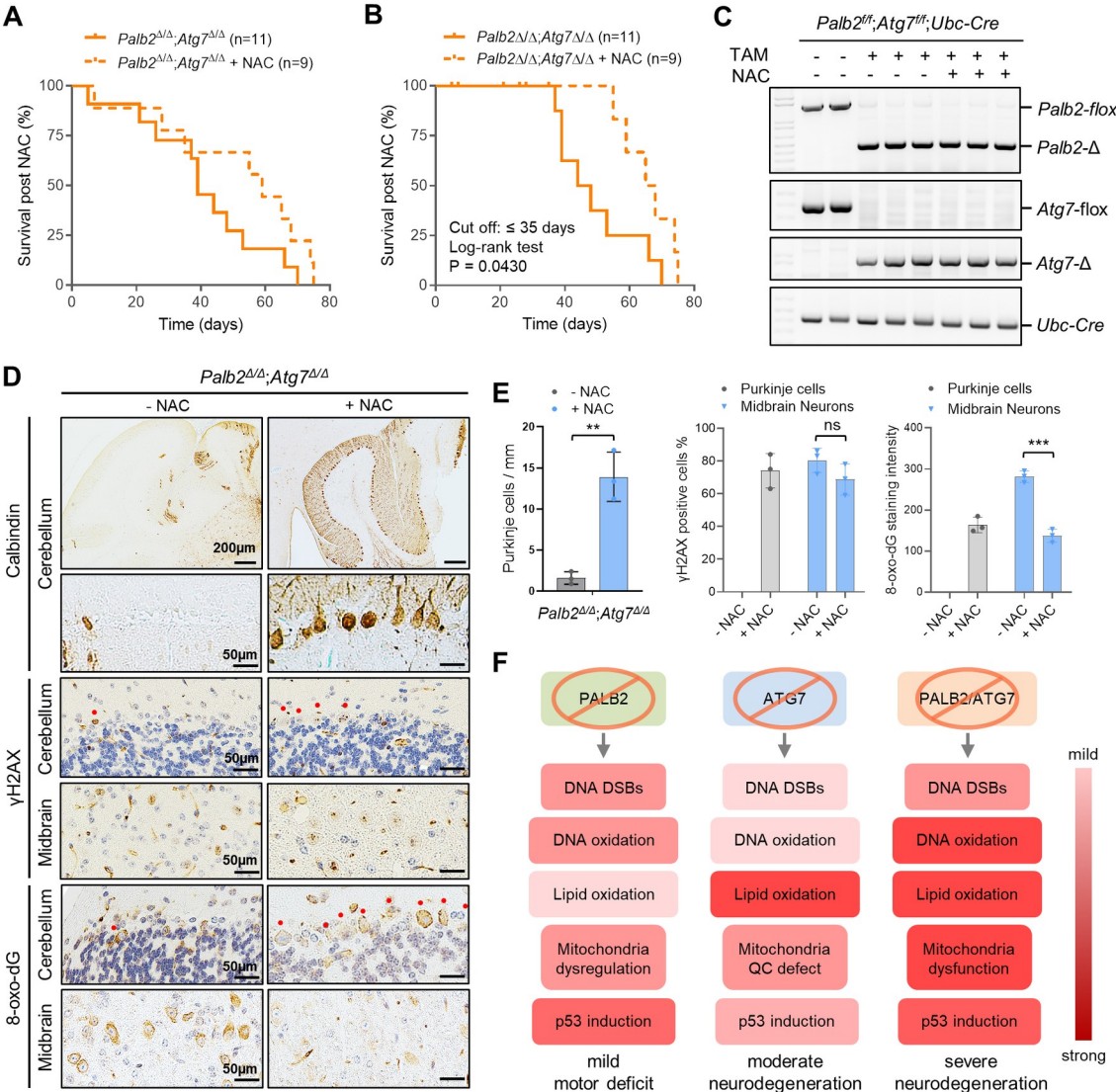

**Fig 8. Partial rescue of *Palb2;Atg7*-WBKO mouse survival by an ROS scavenger and a summary of key events occurring in the brain of the model mice in this study.** (A and B) Survival curves of untreated (n = 11) and NAC-treated (n = 9) *Palb2;Atg7*-WBKO mice. Panel A includes all cause deaths, while in panel B deaths that occurred within the first 35 days were treated as censoring events due to causes unrelated to neurodegeneration. (C) PCR detection of tamoxifen induced, *Ubc-Cre-ERT2*-mediated *Palb2* or *Atg7* deletion in the liver of the mice (note that the whole brain was fixed for all mice). (D) Representative IHC images of Calbindin, γH2AX and 8-oxo-dG in the cerebellum and midbrain of *Palb2;Atg7*-WBKO mice with or without 8 days of NAC treatment. Red dots indicate Purkinje cells. Scale bar = 200 μm or 50 μm as indicated. n = 3 mice per condition. (E) Quantification of Purkinje cell number, γH2AX positivity and 8-oxo-dG staining intensity in the cerebellum/Purkinje cells and midbrain neurons of *Palb2;Atg7*-WBKO mice with or without NAC treatment. n = 3 mice per condition. (F) A summary of the impact of PALB2 and ATG7 loss on DNA damage (DSBs), DNA oxidation, lipid peroxidation, mitochondria and p53 status. Colors from light to dark denote mild to strong effects. See texts for details.

PALB2 is known to promote HR-mediated DSB repair by connecting BRCA1 and BRCA2 [29, 30] and maintain redox homeostasis by protecting NRF2 from KEAP1-mediated repression thereby promoting antioxidant gene expression [31]. In keeping with these known functions, ablation of *Palb2* in the brain led to elevated DNA breaks and DNA oxidation in both Purkinje cells and midbrain neurons (Fig 3A–3D). Although *Palb2*-CKO mice did not show any overt behavioral deficits and neurodegeneration-associated death, mild motor deficits

were observed in these mice (Fig 1F and 1G and 1L). Importantly, our analyses of *Palb2* CKO mouse brains and *PALB2* KO human (DAOY) medulloblastoma cells led to the discovery of a novel role for PALB2 in regulating mitochondrial biogenesis and function. Specifically, loss of PALB2 caused broad upregulation of genes involved in the above processes, accompanied by increased mitochondrial mass in both the mouse brain and human DAOY cells (Figs 3 and 4). This new discovery adds to the repertoire of PALB2 functions and may have implications in not only neurodegeneration but also tumor suppression. The functional relationships between increased mitochondrial biogenesis, decreased mitochondrial respiration and increased ROS upon loss of PALB2 await further investigation.

Interestingly, although ATG7/autophagy and PALB2 both play a role in DNA repair and the maintenance of redox and mitochondrial homeostasis, our detailed analyses of mouse brain tissues and human DAOY cells revealed both similarities and differences between the impact of loss of the two proteins on DNA damage, oxidative stress, and mitochondrial homeostasis, as summarized in Fig 8F. Relatively speaking, in the mouse brain, *Palb2* deletion led to larger increases in DSB formation, DNA oxidation, and mitochondrial mass in Purkinje cells and midbrain neurons, whereas *Atg7* deletion resulted smaller increase of DSB formation and DNA oxidation but stronger lipid peroxidation (Fig 3). Human *PALB2*-KO DAOY cells showed larger increases in mitochondrial mass and higher cellular ROS as detected by DCF-DA, which mostly detects $H_2O_2$, whereas *ATG7*-KO cells showed higher levels of mitochondrial superoxide as detected by MitoSox Red (Fig 5). Compared with either single KO cells, *PALB2*;*ATG7*-DKO cells showed even higher cellular ROS, more severe impairment of mitochondrial function, and more spontaneous cell death. These results suggest that PALB2 and ATG7 regulate different aspects of redox and mitochondrial homeostasis and that the severe phenotypes of the double CKO/WBKO mice and the DKO human cells are due to simultaneous loss of complementary functions of ATG7 and PALB2 in antioxidant defense and mitochondrial homeostasis. As neurons are sensitive to oxidative stress and energy deprivation [55,56], they are also especially vulnerable to combined loss of ATG7/autophagy and PALB2.

p53 has been strongly implicated in neuronal death and neurodegenerative diseases [57,58]. We have also shown that deletion of *Trp53* delays neurodegeneration in *Atg7*-WBKO mice [17]. In the present study, deletion of *Trp53* alleviated the neurodegenerative phenotype of *Palb2*;*Atg7*-CKO mice and significantly prolonged survival (Fig 7), suggesting that p53 plays a key role in causing the neurodegenerative phenotype. Interestingly, *Trp53* deletion did not appear to affect Purkinje cell number in *Atg7*-CKO mice but appeared to restore their number in *Palb2*;*Atg7*-CKO mice to a level similar to that in *Atg7*-CKO mice at 6 weeks of age (Fig 7G), suggesting that the accelerated Purkinje cell loss in *Palb2*;*Atg7*-CKO mice was largely an *Atg7* phenotype worsened by further activation of p53 caused by loss of PALB2. It should be noted that although Purkinje cell loss in *Atg7*-CKO mice may not be caused by p53 activation at 6 weeks, p53 activation may lead to apoptosis in other neurons. Another interesting finding is that although p53 accumulation was stronger in *Palb2*-deleted Purkinje cells than in *Atg7*-deleted cells (Fig 7D and 7E), apoptosis level was much lower in *Palb2*-deleted cells (Fig 2G). This suggests that the accumulated p53 in *Palb2*-deleted Purkinje cells may not be sufficiently active to trigger apoptosis and that p53 in *Atg7*-deleted cells may be more active or act in conjunction with other factors to induce cell death. It is also possible that *Palb2*-null Purkinje cells might be protected from apoptosis by certain prosurvival factors, and this scenario is supported by our recent reports showing NFκB activation in *Palb2* mutant mice and *Palb2*-deleted mammary epithelial cells [59,60].

A key question that remains to be answered is how PALB2 regulates mitochondrial gene expression, biogenesis, and function. These aspects could be regulated either through a common pathway or via separate mechanisms. Given the broad upregulation of many genes

encoding different components of mitochondria, PALB2 may potentially act as a repressor of a master transcription factor that controls mitochondrial gene expression and biogenesis. PGC-1α is a strong candidate for such a factor [61], and its mRNA is indeed upregulated in PALB2-depleted DAOY cells (Fig 4G). PALB2 is a chromatin-bound protein and has been shown to associate with gene promoters and function as a co-activator to regulate their expression [62], although whether it can act as a repressor remains to be seen. To this end, PALB2 may also function together with its partner protein MRG15 [63], which is a key component of the conserved NuA4 chromatin remodeling complex [64]. Another possibility is that the increased mitochondrial biogenesis in PALB2-depleted or KO cells is an adaptive response to impaired mitochondria quality and/or activity, presumably caused by oxidative damage. At the same time, the increased number of functionally impaired mitochondria may also contribute to increased ROS in PALB2 depleted, KO, or mutant cells, in addition to reduced NRF2 activity.

In summary, our in vivo and cell-based analyses revealed a novel function of PALB2 in regulating mitochondrial homeostasis and further established its less studied role in suppressing ROS. Moreover, we showed that loss of PALB2 and ATG7 lead to different degrees of DNA damage, oxidative stress, mitochondrial abnormality, and p53 induction, all of which may contribute to neurodegeneration, although the impact of any moderate difference in each factor remains to be defined. Additionally, we showed that combined deletion of *Palb2* and *Atg7* led to more severe neurodegeneration that was partially rescued by either *Trp53* deletion or antioxidant treatment, suggesting that the severe neurodegenerative phenotypes of *Palb2*;*Atg7* double deletion mice stem mainly from excessive ROS and is at least in part caused by p53-induced neuronal apoptosis. While it is likely that p53 may be induced by DNA damage and oxidative stress caused by loss of PALB2, whether p53 induction and/or activation is also linked to mitochondrial dysregulation in the paradigm used awaits further investigation. The p53-independent component of the cause of neurodegeneration in mice with *Atg7* deletion, whether the autophagy-independent functions of ATG7 are involved, and why *Brca2* deletion ameliorated neurodegeneration in *Atg7*-CKO mice also remain to be elucidated.

## Methods

### Ethics statement

All animal works were approved by the Institutional Animal Care and Use Committee (IACUC) of Rutgers University.

### Mouse models

The *Palb2*$^{flox2-3}$;*Wap-Cre* (*Palb2*-CKO) mice were described previously [27]. They were crossed to strains carrying *Atg7*$^{flox}$ [24], *Brca2*$^{flox11}$ [65], *Trp53*$^{flox2-10}$ [65], or *Ubc-Cre-ERT2* (The Jackson Laboratory) alleles to generate all the genotypes in this study. *Wap-Cre* driven CKO females were all mated to go through two rounds of pregnancy and lactation to induce Cre expression and then monitored for survival and tumor development; males were used for studying neurodegeneration. To generate whole-body somatic *Palb2* and/or *Atg7* knockout in adult mice, 8–10 weeks old males with floxed alleles and *Ubc-Cre-ERT2* were subjected to 5 consecutive daily intraperitoneal injections (unless otherwise specified) of tamoxifen (TAM, 200 μl of 10 mg/ml solution per mouse). A PCR based genotyping method was used to validate *Palb2* deletion as described before [66], and primers used for validation of *Atg7* gene deletion were as follows: forward 5'-aggcagggaggctaaatggt-3', reverse 5'- gggcgccagttaagaacgat-3'. Diagnostic criteria for neurodegeneration-associated death are 1) a mouse has strong motor coordination and balance impairment; 2) loss of more than 15% body weight; and 3) no tumors found upon dissection.

## Motor coordination assessments in mice

Motor coordination was assessed as previously described [67] with minor modifications. For the footprint test, forepaws and hindpaws of the mice were coated with red and green nontoxic paints, respectively. Mice were trained to walk along a 50-cm-long, 10-cm-wide, paper-covered runway (with 10-cm-high walls) into an enclosed box. All mice (5–7 mice per group) were given three runs per day at each time point (6, 9 and 12 weeks of age). A fresh sheet of white paper was placed on the floor of the runway for each run. Footprint patterns were assessed quantitatively by three measurements: stride length, hindbase width and front/hind footprint overlap. For WBKO mice, hind stride length was measured. Beam balance test was performed using a homebuilt experimental setup of a 50 cm-long, 12 mm-diameter round horizontal beam. At each time point (7, 9, 11, 13, 15, 17, 19 and 20 weeks of age), mice (5–7 per group) were first trained to walk on the beam. For data collection, three trials were performed per animal per day. The time the animal remained on the beam and the steps taken without falling were recorded. Both falling and hanging on the beam were counted as falls (a cushioned pad was used to prevent injury). Mice that successfully walked on the beam for 10 steps without falling were assigned 100 points. For mice that could not pass the "10 steps no fall rule", we used a walk and stay combined rule in which their score depended on how many steps they could walk and how long they could stay at the spot where they eventually fell off. In this case, the number of steps taken and length of stay each accounted for half, or 50 points, of the full score, and each step walked or every 6 seconds of stay was given 5 points; these were then added to yield the score for these mice. The final daily beam walking score was the mean score of the three beam-walking trials.

## Whole-body γ-radiation

Mice were retained in a Rodent RadDisk and irradiated using a Gammacell 40 Extractor (MDS Nordion) γ-irradiator at a dose rate of 91.6 cGy/min to a total of dose 10 Gy.

## N-Acetyl-L-Cysteine (NAC) treatment

Eight to ten weeks old *Ubc-Cre-ERT2* model mice were subjected to 4 daily injections of tamoxifen to delete genes of interest and then separated into 2 groups. One group was treated with sterile filtered water supplemented with 1% NAC (Sigma A9165) (pH ~7.4) and the other group was given control water. Water was changed every 2–3 days and the mice were monitored for survival. For histopathology assessment, six 10-week-old female *Palb2^{f/f};Atg7^{f/f};Ubc-CreERT2* mice were subjected to 4-day schedule of TAM injection to delete *Palb2* and *Atg7*. Starting at the first day of TAM administration, three mice were given NAC via drinking water (1 g NAC per 100 ml) and the other three given water without NAC. Mice were euthanized and tissues collected after 8 days of NAC treatment.

## Tissue collection and immunostaining

At indicated time points, mice were anesthetized and perfused first with phosphate-buffered saline (PBS) and then with 4% paraformaldehyde (PFA). Brains were dissected, post fixed in 4% PFA for 24–72 h at 4°C, and then transferred to 70% ethanol prior to further processing. Paraffin-embedded block production and sectioning were conducted by The Histopathology Shared Resources of Rutgers Cancer Institute of New Jersey. IHC was performed on 5 μm sections as described before [68]. Paraffin sections were stained with antibodies against SQSTM1/p62 (1:1000 for IHC and 1:5000 for IF, Abcam, ab91526); Tyrosine Hydroxylase (EP1532Y, 1:300 for IHC and 1:2000 for IF, Abcam, ab137869); Calbindin (EP3478, 1:4000, Abcam, ab108404); γ-H2AX (1:300; Millipore, 05–636), 8-oxo-dG (1:2000; Trevigen); MT-CO1 (1:300,

Abcam, ab14705), 4-HNE (EP1532Y, 1:500, Abcam, ab48506); p53 (CM5, 1:500; Leica Biosystems) and SOD2 (1:200, EMD Millipore, 06–984). Immunostained midsagittal sections were scanned using an Olympus VS120 instrument and images were generated using OlyVIA software for viewing.

Three animals per condition and two sections per animal were used for every quantification. Purkinje cell numbers were measured by total Calbindin positive Purkinje cells per section divided by the total length of Purkinje cell layer measured by OlyVIA. For γ-H2AX staining, in cerebellum Purkinje cells, ≥1 foci/cell were counted as positive; in midbrain neurons, foci number/cell were counted. For quantification of 8-oxo-dG positive ratios in Purkinje cells, both light and dark brown staining were counted as positive. The intensity of staining was recorded as 0, 1, 2, and 3 and the percentage of positive staining was recorded from 0 to 100%. The results of staining were scored using quick (Q) score, which was obtained by multiplying the percentage (P) by the intensity (I) ($Q = P \times I$; maximum = 300). At least 100 Purkinje cells or all Purkinje cells in a brain section (when the total number in the section was less than 100) and at least 50 midbrain neurons/section were counted for each sample. For 4-HNE staining, the intensity of staining was recorded as 0, 1, 2, and 3 and the percentage of positive staining area was recorded from 0 to 100%. For p62 IF staining, images were converted to grayscale and measured by ImageJ on the midsagittal brain sections.

## Cell culture, RNA interference, RNA-Seq analysis and qRT-PCR

DAOY cells were purchased from American Type Culture Collection (ATCC) and cultured in DMEM/F12 medium with 10% heat-inactivated fetal bovine serum (FBS) and 1X penicillin-streptomycin (Pen-Strep) in a humidified incubator with 5% $CO_2$. For *PALB2* knockdown, cells were seeded at $1 \times 10^5$ cells per well in 6 well plates and transfected with siRNAs using Lipofectamine RNAiMax (ThermoFisher, #13778150) following manufacturer's instructions. The final concentration of siRNAs was 8 nM. The sequences of the sense strands of the *PALB2* siRNAs are listed in S2 file. These siRNAs were custom synthesized by Sigma Genosys. Control AS siRNA was purchased from Qiagen (AllStars, #1027281), other nonspecific siRNAs were synthesized by Sigma Genosys. Three days after transfection, total RNA was extracted using RNeasy Plus Mini Kit (#74134, QIAGEN). Library preparation, RNA-seq and bioinformatics analysis was performed at Novogene (Beijing, China). Briefly, the sequencing reads were aligned to human genome (version Hg19) and the resulting binary alignment (BAM) files were used to calculate the gene counts that represent total number of sequencing reads aligned to a gene. DEGs between control and *PALB2* siRNA-treated samples were selected based on fold change of ≥1.5 for upregulated and ≤0.67 for downregulated genes, Cutoff P value is 0.05. Top 50 upregulated DEGs and top 50 downregulated DEGs were depicted as heatmap, GSEA was performed using GSEA software (version 4.0.0; https://www.gsea-msigdb.org) [69].

For qRT-PCR, total RNA was extracted and used for poly-T-based reverse-transcription (SuperScript III First-Strand Synthesis System, 18080051, Invitrogen) according to manufacturer's protocol. qRT-PCR was performed using an ABI-7300 sequence detection system using SYBR Green qPCR master mix (Applied Biosystems A25742). Each measurement was performed in duplicate and expression levels of *ACTB* were used for normalization. Only primer pairs resulting in a single peak in the melting curve analysis were used. Oligonucleotides used for qPCR are listed in S2 File.

## CRISPR/Cas9 knock out of *PALB2* and *ATG7*

pSpCas9-2A-GFP (PD1301) V2.0 constructs containing gRNAs targeting *PALB2* and pSpCas9 (BB)-2A-Puro (PX459) V2.0 constructs containing gRNAs targeting *ATG7* were transfected

into DAOY cells using X-tremeGENE 9 (Roche, Mannheim, Germany). Single cell clones were isolated by puromycin selection or fluorescence activated cell sorting (FACS). Clones were first assayed by western blotting for complete loss of protein expression, and genomic DNA of candidate clones were then sequenced to verify the disruption of the genes. The target sites and sequences of guide RNAs were listed in S2 File. Cell doubling time was calculated by the following formula: $r = \frac{\ln(N(t)/N0)}{t}$ Where N(t) = the number of cells at time t; N0 = the number of cells at time 0; r = growth rate; t = time (usually in hours) *doubling time* $= \frac{\ln2}{r}$

## Western blotting

Cells were lysed with NETNG-400 (400mM NaCl, 1 mM EDTA, 20 mM Tris-HCl [pH7.5], 0.5% Nonidet P-40, and 10% glycerol) with Complete protease inhibitor cocktail (Roche). Tissues were homogenized in the same buffer with a TissueRuptor (Qiagen) followed by sonication. Samples were resolved on 4–12% or 4–20% Tris-glycine SDS-polyacrylamide gels and transferred to a nitrocellulose membrane (0.45 micron, Bio-Rad) overnight at 4°C. Resolved proteins were detected following standard procedures using the following antibodies: ATG7 (1:2000; Sigma A2856), LC3 (1:1500; Novus Biologicals NB600-1384), p62 (1:10000; Abcam, ab109012), PALB2 M11 (1:2000, homemade), SOD2 (1:2000, EMD Millipore, 06–984), MT-CO1 (1:2000, Abcam, ab14705), MT-CO2 (1:1000, Abcam, ab198286), GAPDH (1:1000; Santa Cruz Biotechnology sc-365062), and β-Actin (1:5000; Sigma A1978). Blots were developed using Immobilon Western Chemiluminescent HRP Substrate (EMD Millipore).

## GSH/GSSG measurement

In brief, brain tissues were collected from the *Palb2*, *Atg7*, and *Palb2;Atg7* WBKO mice at 4 weeks post tamoxifen treatment and snap frozen in liquid N2. Frozen tissue samples were weighed (~20–30 mg) and ground in liquid N2 using Cryomill (Retsch). The powdered samples were then mixed with methanol: acetonitrile: water (40:40:20) solution containing 0.5% formic acid, followed by 10 sec vortexing, 10 min incubation on ice and centrifugation at 16,000 x g at 4°C. The supernatants were collected, neutralized with 50 μL/mL of 15% ammonium bicarbonate, and then centrifuged at 16,000 x g for 10 min. The final supernatants were transferred to LC–MS autosampler vials and sent for LC-MS analysis which was done using Q Exactive Plus hybrid quadrupole orbitrap mass spectrometer (ThermoFisher). Raw data analysis was done using *Maven* v.707. 4 [70]. GSH and GSSG contents in the brain were measured as part of a comprehensive metabolomics analysis using liquid chromatography-mass spectrometry (LC-MS).

## Terminal deoxynucleotidyltransferase dUTP nick end labeling (TUNEL) assay

Staining was performed on 5 μm sections using the DeadEnd Fluorometric TUNEL System (Promega) according to manufacturer's instructions, and all apoptotic cells in the cerebellum of each midsagittal section were counted.

## Measurements of ROS, mitochondrial mass, mitochondrial membrane potential

Cells were washed with PBS and then incubated at dark for 30 min in phenol red–free DMEM with 10% FBS and 100 μmol/L 2',7'-dichlorofluorescein diacetate (DCF-DA; D6883, Sigma) or 2.5 μmol/L MitoSOX Red (M36008, Invitrogen) for ROS measurement [60], 200 nmol/L of MitoTracker Green FM (M7512, Invitrogen) for mitochondrial mass measurement, and 200

nmol/L of MitoTracker Red CMXRos (M7512, Invitrogen) for mitochondrial membrane potential measurement. After incubation, cells were trypsinized, spun down, and resuspended in PBS at a density of approximately $2 \times 10^5$ cells per mL. Signals were analyzed by flow cytometry.

### Cell death measurement

Three wells of the above cells were directly trypsinized and used to measure apoptosis and necrosis using the FITC Annexin V Apoptosis Detection Kit with PI (BioLegend, #640914) according to manufacturer's instructions.

### Assessment of oxygen consumption rate (OCR)

OCR of control and KO DAOY cells were measured using a Seahorse Biosciences extracellular flux analyzer (XF24). Cells were seeded at $2.0 \times 10^4$ cells per well in the XF24 plates overnight prior to XF assay. Real-time OCR measurements were performed in DMEM/F12 with 2 mM glutamine, or 1 mM dimethylα-KG for 3 h, and measurements were taken every 15 min. Relative OCR (percentage) was normalized to the 0-min time point.

### Statistical analyses

Comparisons of survival curves were made using the log-rank test, and unpaired two-sided Student's t test was used for column analysis with GraphPad Prism 8.4.3. All error bars represent standard error of the mean (SEM). Significance is denoted as follows: ns, $p \geq 0.05$; *, $p < 0.05$; **, $p < 0.01$. ***, $p < 0.001$.

## Supporting information

**S1 Fig. Hind-leg clasping reflex test of *Palb2*, *Atg7* and *Palb2;Atg7* CKO mice.**
(PDF)

**S2 Fig. Detection of *Palb2* and/or *Atg7* deletion in *Wap-cre* or *Ubc-Cre-ERT2* model mice.**
(PDF)

**S3 Fig. Expression of p62/SQSTM1 and tyrosine hydroxylase (TH) in the brains of *Palb2*, *Atg7* and *Palb2;Atg7* CKO mice.**
(PDF)

**S4 Fig. IHC staining of p62/SQSTM1 and TH in cross sections of the coronal of *Palb2*, *Atg7* and *Palb2;Atg7* CKO mice.**
(PDF)

**S5 Fig. IHC analysis of markers of DNA damage, oxidative stress and mitochondria in Purkinje cells of *Palb2*, *Atg7* and *Palb2;Atg7* CKO mice.**
(PDF)

**S6 Fig. Markers of DNA damage, oxidative stress and mitochondria in *Palb2*, *Atg7* and *Palb2;Atg7* WBKO mice and the sensitivity of the mice to gamma radiation.**
(PDF)

**S7 Fig. Screening of *PALB2*-KO, *ATG7*-KO and *PALB2;ATG7*-DKO DAOY cell clones and measurements of mitochondria in the clones.**
(PDF)

**S1 File. RNA-seq data and analysis**
(XLSX)

**S2 File. Sequences siRNAs, sgRNA target sites and primers used for qRT-PCR.**
(XLSX)

**S3 File. Numerical data for Fig 1A, 1B, 1C, 1E, 1F, 1G, 1H, 1I, 1J, 1K and 1L.**
(XLSX)

**S4 File. Numerical data for Fig 2B, 2D, 2F and 2H.**
(XLSX)

**S5 File. Numerical data for Fig 3B, 3D, 3F, 3G, 3I and 3K.**
(XLSX)

**S6 File. Numerical data for Fig 4G and 4J.**
(XLSX)

**S7 File. Numerical data for Fig 5B, 5E, 5G, 5J and 5K.**
(XLSX)

**S8 File. Numerical data for Fig 5H and 5I.**
(XLSX)

**S9 File. Numerical date for Fig 6A, 6B, 6C, 6D, 6E, 6F, 6H, 6I and 6J.**
(XLSX)

**S10 File. Numerical data for Fig 7A, 7B, 7C, 7E, 7G, 7H, 7I, 7J and 7K.**
(XLSX)

**S11 File. Numerical data for Fig 8A, 8B and 8E.**
(XLSX)

**S12 File. Numerical data for S5B, S5D, S5F Fig.**
(XLSX)

**S13 File. Numerical data for S6B Fig.**
(XLSX)

## Acknowledgments

We thank Ms. Zhixian Hu, Dr. Xiaoyang Su, Dr. Gabriele Vincelli and Dr. Saurabh Laddha for their help in mouse breeding, metabolomics analysis, CRISPR cell line generation, and RNA-seq data analysis, respectively.

## Author Contributions

**Conceptualization:** Eileen White, Bing Xia.

**Data curation:** Yanying Huo, Akshada Sawant.

**Formal analysis:** Yanying Huo, Akshada Sawant, Hui Ma.

**Funding acquisition:** M. Maral Mouradian, Eileen White, Bing Xia.

**Investigation:** Yanying Huo, Akshada Sawant, Yongmei Tan, Amar H Mahdi, Tao Li, Jake Coleman.

**Methodology:** Yanying Huo, Akshada Sawant, Vrushank Bhatt, Run Yan, Jessie Yanxiang Guo.

**Project administration:** Bing Xia.

**Resources:** Eileen White, Bing Xia.

**Supervision:** M. Maral Mouradian, Eileen White, Bing Xia.

**Writing – original draft:** Yanying Huo.

**Writing – review & editing:** Akshada Sawant, Hui Ma, Cheryl F Dreyfus, Jessie Yanxiang Guo, M. Maral Mouradian, Eileen White, Bing Xia.

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
