## [Decision Letter · Decision Letter 0]

11 Oct 2021

Dear Dr Bing Xia,

Thank you very much for submitting your Research Article entitled 'Tumor suppressor PALB2 maintains redox and mitochondrial homeostasis in the brain and cooperates with ATG7/autophagy to suppress p53 dependent neurodegeneration' to PLOS Genetics.

The manuscript was fully evaluated at the editorial level and by independent peer reviewers. The reviewers appreciated the attention to an important problem, and raised some substantial concerns about the current manuscript. Based on the reviews, we will not be able to accept this version of the manuscript, but we would be willing to review a much-revised version. We cannot, of course, promise publication at that time.

Should you decide to revise the manuscript for further consideration here, your revisions should address the specific points made by each reviewer (especially the concerns from reviewers 2 and 3). We will also require a detailed list of your responses to the review comments and a description of the changes you have made in the manuscript.

If you decide to revise the manuscript for further consideration at PLOS Genetics, please aim to resubmit within the next 60 days, unless it will take extra time to address the concerns of the reviewers, in which case we would appreciate an expected resubmission date by email to plosgenetics@plos.org.

[LINK]

We are sorry that we cannot be more positive about your manuscript at this stage. Please do not hesitate to contact us if you have any concerns or questions.

Yours sincerely,

Gabriel Balmus, D.V.M., Ph.D.

Guest Editor

PLOS Genetics

Gregory Barsh

Editor-in-Chief

PLOS Genetics

Reviewer's Responses to Questions

**Comments to the Authors:**

Reviewer #1: In this manuscript, Huo et al reported the role of PALB2 (a tumor suppressor that is known to be essential for DNA repair, redox homeostasis and breast cancer suppression) and ATG7/autophagy in the suppression of neurodegeneration in mice. Using two types of Cre mice (Wap-Cre [leaky expressing Cre in neurons] and Ubc-Cre-ERT2), the authors found that co-deletion of Palb2 and Atg7 causes severer neurodegeneration and earlier lethality compared to single KO mice. Pathological studies revealed the enhanced DNA damage, oxidative stress, mitochondrial dysfunction and apoptotic cell death, especially in Purkinje cells, in double KO mice compared to single Ko mice, suggesting that Palb2 and Atg7 may synergistically function to suppress these abnormalities. The cellular phenotypes were recapitulated in double KO medulloblastoma DAOY cell lines. Mechanistically, the authors demonstrated that further loss of p53, but not Braca2 (a critical factor for DNA repair) partially rescued the phenotypes (lethality, Purkinje cell loss, and abnormal behaviors) of double KO mice at least to the levels of Atg7 KO mice. Survival rates of double KO mice were also partially rescued by treatment with NAC, a ROS scavenger, suggesting that high levels of oxidative stress rather than increased DNA damage contributed to the lethality of double KO mice. Taken together, the authors concluded that PALB2 maintains redox and mitochondrial homeostasis in the brain cooperatively with Atg7.

Although the critical role of autophagy in protecting against oxidative stresses is already well established and it is easy to assume that deletion of two different and important pathways can cause severe phenotypes, this study for the first time demonstrates the genetic interactions between PALB2 and autophagy in the brain. To support their conclusion, however, there remain some concerns that need to be clarified.

1. In Figure 7, the causal relationships between the higher oxidative stress and the severer phenotypes of double KO mice are unclear. The authors should confirm the effect of NAC on neurodegeneration, oxidative stress, and DNA damages by histological analyses. The effect of NAC could also be checked using DAOY cell lines.

2. The authors should discuss or provide further evidence about the following points: (1) how Palb2 regulates redox and mitochondrial homeostasis, (2) how p53 causes severer phenotypes, (3) whether autophagy-independent functions of Atg7 are involved or not.

Reviewer #2: PALB2 is a tumor suppressor protein associated with familial breast cancer and is implicated in DNA repair and redox homeostasis. In the present study Huo et al. report that Palb2 deletion in the mice brain (driven by Wap-Cre) or in the whole-body (driven by Ubc-Cre-ERT2) lead to motor deficits and its co-deletion with the essential autophagy gene Atg7 accelerates and exacerbates neurodegeneration that was initially induced by Atg7 deletion. Such a co-deletion led to loss of Purkinje cells (controller neurons of motor coordination and balance) accompanied by increased DNA damage, oxidative stress and mitochondrial dysfunction. The effect of mitochondrial dysfunction and oxidative stress was verified in PALB2 KO human DAOY medulloblastoma cells. To rull out the possibility that the DNA damage activity of PALB2 is involved in the authors have generated a double knockout mouse in which BRCA2 was deleted together with ATG7. Such a deletion led to a delay in neurodegeneration induced by ATG7 and therefore they concluded that the exacerbating effect of neurodegeneration in Palb2;Atg7-CKO mice is not caused by the loss of DNA repair function of PALB2. Finally, a triple KO mouse was created in which p53, Palb2 and Atg7 were co-deleted, leading to partial recovery the neurodegeneration phenotype. This led the authors to conclude that the overall neurodegeneration caused by the lack of Atg7 and PALB2 is under the control of p53.

While the body of work invested in this study is well appreciated, the main conclusions and the authors’ model are not fully supported by the data, which mostly remain too preliminary. Almost every claim the authors are making requires additional complementary experimental evidence which are missing throughout. For example, the new role for PALB2 in the regulation of mitochondrial biogenesis and function and that this role leads to exacerbating effect of neurodegeneration in the absence of ATG7 should be better characterized. The authors ignore the role of autophagy in the regulation of DNA damage which seems important here.

This is also relevant to the conclusion that p53 indeed plays a role in this process. Is it a direct involvement? Which of the downstream factors are needed here? These and many more questions remain unanswered.

Additional comments:

1. It is well characterized that the loss of ATG7 leads to neurodegeneration, but the authors indicated that :”Unexpectedly, Atg7-CKO mice showed progressive motor deficit….suggestive of cerebellar involvement”.

2. The authors claim that PALB2 exacerbates neurodegeneration induced by ATG7 loss, but they did not test any markers of neurodegeneration additionally to Purkinje cells survival.

3. Likewise, the requirement of PALB2 for mitochondrial homeostasis was hardly investigated and the presented results are not convincing. Moreover, the claim that “PALB2 leads to increased mitochondrial biogenesis” was not explored at all (real time PCR of mitochondrial genes or any other method).

4. The effect of p53 deletion was not deeply explored and its role in the partial recovery of the Palb2;Atg7;Trp53-CKO mice survival remains mostly not clear.

5. Figure 1A – survival discrepancy of Atg7-CKO mice between the text (604 days) and the figure (596 days).

6. The Atg7-CKO mice showed limb clasping reflexes when suspended by the tails, but for an unclear reason that data was not shown What about Palb2;Atg7-CKO?

7. Figure 2A – the staining for tyrosine hydrolase (TH) a marker of dopaminergic neurons does not contribute to this figure and may be presented in supplementary figures. Instead, it may be more productive to co-stain any other neurodegenerative markers together with p62.

8. Figure 2D – the TUNEL staining for 8 weeks should be presented to be in line with 2E.

9. Figure 3 – all images for all time points should be presented (at least in supplementary material). The high signals in Atg7-CKO and Palb2;Atg7-CKO after six weeks described in panels 3B and 3D is not clear given that apoptosis already occurred. The explanation in the text is unclear.

10. Figure 4F and 4G – the measurements of total mitochondria and mitochondrial membrane potential only by mitotrackers immunofluorescence are not sufficient, should be validated at least by FACS.

11. Figure 4H and 4I – the OCR measured by seahorse represents only 4 cell lines, but the mitochondria parameters of that measurements (I) show 8 cell lines. These two panels should be in line. In addition, the authors claim that the double knockout DAOY medulloblastoma cells showed strong defect in mitochondrial respiration bue to spontaneous death of DKO, but may be 100 minutes are not enough for DKO necrosis\\apoptosis and they just switch to glycolysis?

12. Figure 6 – a plot of cause of death (%) is missing.

13. Figure 7A and 7B – the recovery phenotype by ROS scavenger is more relevant to figure 3. Panels C and D are not contributing since the model is too preliminary and requires additional investigation.

14. The authors decided to start the results section trying to “…to further examine the role of autophagy in mammary tumor development…” but moved to determine neurodegenerative effects since they used C57BL\\6 genetic background which is resistant to mammary tumorigenesis.

15. The discussion section is mostly a rewriting of the results and it suffers from lack of relevant references.

Reviewer #3: In this manuscript, the role of the tumour suppressor Palb2 on neurodegeneration is studied, especially regarding its loss in synergy with the loss of the essential for autophagy gene Atg7. It is shown that loss of PALB2 accelerates and exacerbates cerebellar Purkinje cell degeneration observed in Atg7 KO animals resulting in motor function deficits and reduced survival, and that this PALB2 action is exerted at least partially through p53 activity. The data reveal a role of PALB2 on mitochondrial function and reinforce existing data on the role of PALB2 in redox homeostasis, which could contribute to neurodegeneration observed. Partial rescue of survival after treatment of double KO animals with a ROS scavenger confirms the role of ROS accumulation in reduced survival. Moreover, it further suggests disturbed redox homeostasis as a mechanism contributing to the exacerbation of neuronal degeneration observed in Atg7 KO animals, upon PALB2 loss. This is a manuscript with interesting findings, and a wide set of, in large, convincing experiments. Nevertheless it suffers in results presentation, as well as in text structure and phrasing. Moreover, a small set of the data presented should be further analyzed. We suggest the following changes before publication:

Major

1. The results and figures sections have to be more carefully crafted, as the following mistakes are recurrent:

a. Describe with clear distinction between observations of staining intensity and descriptions of the quantification of IHC stainings. Lack of clear separation between the two is a recurrent problem throughout the results text (e.g. results paragraph “Requirement of PALB2 for mitochondrial homeostasis in the brain”).

b. In results, page 7, authors describe differences observed in p62 IF stainings, without mentioning the age in which these differences are observed. This happens again during the results regarding 4-HNE staining (results, pages 9-10). Mouse ages should be found on every result description throughout the results section.

c. While one finds results from experiments on multiple mouse ages in the relevant figures, these results are not described in the text, e.g. description of the results of in vivo MT-CO1 stainings for weeks 6-10 is missing from the text (results page 10), while description of the results from quantification of the SOD2 staining is completely missing from the same paragraph. Description of the p53+ Purkinje cell quantification from results, page 14, for ages 8-10 is also not provided. These and all results presented in the figures should be described in more detail in the results text.

d. Often, in the figures presenting data, quantification results from several mice ages are found in the graphs, but no representative images are shown for all the mice ages (2B-C, 2D-E, 3A-B, 3C-D, 3G-H, 3I-J, 6D-E). It would be more informative if images from all ages presented in the graphs would also be shown in the figures.

e. Moreover, the main text references to the figures should be accurate. For example, in page 14, the reference to Figures 6F and G should change to refer to only figure 6G.

2. In results, page 5, T50 of the Atg7-CKO mice is reported to be 604 in the text and 546 in Figure 1A. Please correct this discrepancy.

3. Supplementary figure 1A should be better explained. Why do Palb2f/f;Atg7f/f Ubc-Cre and Atg7f/f Ubc-Cre mice have both Atg7-Δ and Atg7-flox bands? Is there no way of telling through PCR wt Palb2 from flox Palb2 allele? Also, the Atg7-wt bands are not very convincing.

4. Regarding p62 IF, the difference is obvious from the images in the age of 10 week old mice, but not in 6 week old mice (if any). In general images and specific brain areas depicting p62 IF should be in higher magnification.

5. Regarding tyrosine hydroxylase IF, a higher magnification of the substantia nigra, or quantification of the positive cells in the substantia nigra would be necessary.

6. Regarding 4-HNE staining. This is a result that is found useful by the authors in data interpretation later on in the manuscript; it should, thus, be more substantiated with quantification, and not based only on observation.

7. Statistical analysis for results in several images is missing (3H, 3J, 5D, 6E, 6I).

8. Discussing experiments regarding treatment of the Ubc-Cre-ERT2 model animals with N-Acetyl-Cysteine, you find that NAC treatment prolongs survival of Palb2;Atg7-WBKO mice and suggest that NAC protects against late death cause, which probably is neurodegeneration. Are there motor coordination data from NAC-treated animals such as delay of neurological symptoms onset, or IHC data regarding Purkinje cell survival supporting this suggestion?

9. In the materials and methods section authors describe a preparation of tissues for electron microscopy to measure mitochondrial mass, but relevant results are not discussed in the results section and in figures. It would be greatly adding to the manuscript to present this data.

10. In the materials and methods section authors mention that they used male KO mice for neurodegeneration and female for carcinogenesis. Please explain your thinking in more detail. Where no female mice used for any neurodegeneration study and motor assessment?

11. In results, page 16, authors mention: “Our studies also establish PALB2 as a suppressor of neurodegeneration in its own right, representing another new function of PALB2”. As by itself loss of PALB2 does not affect neurodegeneration in a great extent, this statement while not false, should be downtoned.

12. In the results concerning BRCA2 loss, page 13, authors mention: “As shown in Figure 5F-I, γH2AX positivity in Purkinje cells of Brca2-CKO mice was slightly lower than that in Palb2-CKO mice but still much higher than that in control mice; in contrast, their 8-oxo-dG and MT-CO1 staining signals were substantially weaker than those in Palb2- deleted Purkinje cells“. In this case, the results show that a statistically significant smaller number of Purkinje cells from Brc2-CKO mice are positive for γH2AX and 8-oxo-dG in comparison with the Palb2-CKO mice, while a statistically significant higher number of Purkinje cells from Brc2-CKO mice are positive for γH2AX and 8-oxo-dG in comparison to the Wap-cre mice, and this is similar for both markers. Thus, a comparison of positivity levels between these markers cannot lead to safe conclusions. The same goes for MT-CO1 marker. Also, comparing the results coming from staining intensity quantification (MT-CO1) to results coming from cell positivity measurements seems to be a stretch. Please rephrase.

13. Annexin V assay is not described in the methods, the data is not but mentioned in the results text, and there is no statistics shown. Has this experiment been duplicated? Please provide all relevant information in the methods section, and explain in detail the data in the results section.

14. Accurate number of animals used, as well as times of experiment repeated should be mentioned in detail for all experiments and each group and time point. Also, it is recommended to have at least three animals in each group examined so statistical analyses results are robust.

15. Add scale bars to all microscopy images and indicate scale bar length in relevant figure legends.

16. In general, figure legends have minimal information. They should be more descriptive.

Minor

1. Colour problems in bars of graphs are found in figures 3B, 3D, 5G, 5H, 6E, 6I, 6J.

2. In introduction, page 4, the phrase “In the current study, we aimed to co-delete Palb2 and Atg7 in the mammary gland using Cre recombinase driven by the promoter of whey acidic protein (Wap-Cre), to further study the role of autophagy in PALB2-associated breast cancer” could be misleading. Please rephrase, as breast cancer is not the main focus of the current study.

3. Missing in materials and methods: TH, p62, 4-HNE IHC staining information, Annexin V assay information. No γ-radiation methods are given. No cell doubling time measuring method for cell culture is provided.

4. Regarding results of graphs in figures 7A and 7B, please explain them in more detail in the results text.

5. Indicate whether the stride result for WBKO animals presented in figure 1L represents fore or hind or both types of paws.

6. Please reconsider the use of the word “condensed signal” to describe staining signal, in results page 10.

7. The last sentence of paragraph “Requirement of PALB2 for mitochondrial homeostasis in the brain” describing MT-CO1 signal intensity for Palb2Δ/Δ mice should be moved right after the MT-CO1 data from the Atg7-CKO and Palb2;Atg7 double CKO mice in the same paragraph.

8. In results, page 11, authors mention: “In keeping with our in vivo findings, MT-CO1, MT-CO2 and SOD2 were all increased in PALB2-KO, ATG7-KO and PALB2;ATG7-DKO cells, implying increased mitochondrial mass upon loss of either PALB2 or ATG7”

a. Please rephrase, as there are no data regarding MT-CO2 in the in vivo findings

b. Please add reference correlating markers MT-CO1 and SOD2 with increased mitochondrial mass.

9. In results, page 15, change “rather than increased DNA damage, in the brain” to rather than increased DNA damage, in the cerebellum”.

10. Add the full term of an abbreviation when used for the first time in the main text.

11. In introduction, page 3, and reference list, correct reference (Collaborators, 2019).

12. Remove “n=4” from fig. 1E.

13. Correct I to L where appropriate in the legend of Figure 1.

14. Correct “tyrosine hydrolase” to “tyrosine hydroxylase” in the relevant results section.

15. Correct spelling and grammar mistakes such as “to expresses Cre” to “to express Cre”, “LC3-I o LC3-II” to “LC3-I to LC3-II”, “Supplementray” to “Supplementary”, “repiration” to “respiration” in results section, “neurodegenetation” to “neurodegeneration”, “acvtivation” to “activation” in the discussion section, e.t.c. Also “while” to “which” in discussion (page 16, the sentence regarding DCF-DA).

16. Change “While suppressing neurodegeneration…” to “Besides suppressing neurodegeneration…” in the introduction.

**Have all data underlying the figures and results presented in the manuscript been provided?**

Reviewer #1: Yes

Reviewer #2: Yes

Reviewer #3: Yes

PLOS authors have the option to publish the peer review history of their article (what does this mean?). If published, this will include your full peer review and any attached files.

Reviewer #1: No

Reviewer #2: No

Reviewer #3: No

---

## [Decision Letter · Decision Letter 1]

9 Mar 2022

Dear Dr Xia,

We are pleased to inform you that your manuscript entitled "Tumor suppressor PALB2 maintains redox and mitochondrial homeostasis in the brain and cooperates with ATG7/autophagy to suppress neurodegeneration" has been editorially accepted for publication in PLOS Genetics. Congratulations!

Yours sincerely,

Gabriel Balmus, D.V.M., Ph.D.

Guest Editor

PLOS Genetics

Gregory Barsh

Editor-in-Chief

PLOS Genetics

Comments from the reviewers (if applicable):

Reviewer's Responses to Questions

**Comments to the Authors:**

Reviewer #1: The authors have addressed all my concerns and therefore I support the publication.

Reviewer #2: None

Reviewer #3: The authors have properly addressed all issues raised and therefore I recommend publication.

**Have all data underlying the figures and results presented in the manuscript been provided?**

Reviewer #1: None

Reviewer #2: Yes

Reviewer #3: Yes

PLOS authors have the option to publish the peer review history of their article (what does this mean?). If published, this will include your full peer review and any attached files.

Reviewer #1: No

Reviewer #2: No

Reviewer #3: No

**Data Deposition**

http://datadryad.org/submit?journalID=pgenetics&manu=PGENETICS-D-21-01092R1

**Press Queries**

---

## [Editor Report · Acceptance letter]

4 Apr 2022

PGENETICS-D-21-01092R1 

Tumor suppressor PALB2 maintains redox and mitochondrial homeostasis in the brain and cooperates with ATG7/autophagy to suppress neurodegeneration 

Dear Dr Xia, 

We are pleased to inform you that your manuscript entitled "Tumor suppressor PALB2 maintains redox and mitochondrial homeostasis in the brain and cooperates with ATG7/autophagy to suppress neurodegeneration" has been formally accepted for publication in PLOS Genetics! Your manuscript is now with our production department and you will be notified of the publication date in due course.

With kind regards,

Livia Horvath

PLOS Genetics

On behalf of:
